# Finite-Time Logarithmic Bayes Regret Upper Bounds

**Alexia Atsidakou**
University of Texas, Austin

**Branislav Kveton**
AWS AI Labs*

**Sumeet Katariya**
Amazon

**Constantine Caramanis**
University of Texas, Austin

**Sujay Sanghavi**
University of Texas, Austin / Amazon

## Abstract

We derive the first finite-time logarithmic Bayes regret upper bounds for Bayesian bandits. In a multi-armed bandit, we obtain $O(c_\Delta \log n)$ and $O(c_h \log^2 n)$ upper bounds for an upper confidence bound algorithm, where $c_h$ and $c_\Delta$ are constants depending on the prior distribution and the gaps of bandit instances sampled from it, respectively. The latter bound asymptotically matches the lower bound of Lai (1987). Our proofs are a major technical departure from prior works, while being simple and general. To show the generality of our techniques, we apply them to linear bandits. Our results provide insights on the value of prior in the Bayesian setting, both in the objective and as a side information given to the learner. They significantly improve upon existing $\tilde{O}(\sqrt{n})$ bounds, which have become standard in the literature despite the logarithmic lower bound of Lai (1987).

## 1 Introduction

A *stochastic multi-armed bandit* [Lai and Robbins, 1985, Auer et al., 2002, Lattimore and Szepesvari, 2019] is an online learning problem where a *learner* sequentially interacts with an environment over $n$ rounds. In each round, the learner takes an *action* and receives its *stochastic reward*. The goal of the learner is to maximize its expected cumulative reward over $n$ rounds. The mean rewards of the actions are unknown *a priori* but can be learned by taking the actions. Therefore, the learner faces the *exploration-exploitation dilemma*: *explore*, and learn more about the actions; or *exploit*, and take the action with the highest estimated reward. Bandits have been successfully applied to problems where uncertainty modeling and subsequent adaptation are beneficial. One example are recommender systems [Li et al., 2010, Zhao et al., 2013, Kawale et al., 2015, Li et al., 2016], where the actions are recommended items and their rewards are clicks. Another example is hyper-parameter optimization [Li et al., 2018], where the actions are values of the optimized parameters and their reward is the optimized metric.

Cumulative regret minimization in stochastic bandits has been traditionally studied in two settings: frequentist [Lai and Robbins, 1985, Auer et al., 2002, Abbasi-Yadkori et al., 2011] and Bayesian [Gittins, 1979, Tsitsiklis, 1994, Lai, 1987, Russo and Van Roy, 2014, Russo et al., 2018]. In the frequentist setting, the learner minimizes the regret with respect to a fixed unknown bandit instance. In the Bayesian setting, the learner minimizes the average regret with respect to bandit instances drawn from a prior distribution. The instance is unknown but the learner knows its prior distribution. The Bayesian setting allows surprisingly simple and insightful analyses of Thompson sampling. One fundamental result in this setting is that linear Thompson sampling [Russo and Van Roy, 2014] has a comparable regret bound to `LinUCB` in the frequentist setting [Abbasi-Yadkori et al., 2011, Agrawal and Goyal, 2013, Abeille and Lazaric, 2017]. Moreover, many recent meta- and multi-task bandit works [Bastani et al., 2019, Kveton et al., 2021, Basu et al., 2021, Simchowitz et al., 2021, Wang et al., 2021, Hong et al., 2022, Aouali et al., 2023] adopt the Bayes regret to analyze the stochastic

structure of their problems, that the bandit tasks are similar because their parameters are sampled i.i.d. from a task distribution.

Many bandit algorithms have frequentist regret bounds that match a lower bound. As an example, in a $K$-armed bandit with the minimum gap $\Delta$ and horizon $n$, the gap-dependent $O(K\Delta^{-1}\log n)$ regret bound of UCB1 [Auer et al., 2002] matches the gap-dependent $\Omega(K\Delta^{-1}\log n)$ lower bound of Lai and Robbins [1985]. Moreover, the gap-free $\tilde{O}(\sqrt{Kn})$ regret bound of UCB1 matches, up to logarithmic factors, the gap-free $\Omega(\sqrt{Kn})$ lower bound of Auer et al. [1995]. The extra logarithmic factor in the $\tilde{O}(\sqrt{Kn})$ bound can be eliminated by modifying UCB1 [Audibert and Bubeck, 2009]. In contrast, and despite the popularity of the model, matching upper and lower bounds mostly do not exist in the Bayesian setting. Specifically, Lai [1987] proved *asymptotic* $c_h \log^2 n$ upper and lower bounds, where $c_h$ is a prior-dependent constant. However, all recent Bayes regret bounds are $\tilde{O}(\sqrt{n})$ [Russo and Van Roy, 2014, 2016, Lu and Van Roy, 2019, Hong et al., 2020, Kveton et al., 2021]. This leaves open the question of finite-time logarithmic regret bounds in the Bayesian setting.

In this work, we answer this question positively and make the following contributions:

1. We derive the first finite-time logarithmic Bayes regret upper bounds for a Bayesian *upper confidence bound (UCB)* algorithm. The bounds are $O(c_\Delta \log n)$ and $O(c_h \log^2 n)$, where $c_h$ and $c_\Delta$ are constants depending on the prior distribution $h$ and the gaps of random bandit instances sampled from $h$, respectively. The latter matches the lower bound of Lai [1987] asymptotically. When compared to prior $\tilde{O}(\sqrt{n})$ bounds, we better characterize low-regret regimes, where the random gaps are large.

2. To show the value of prior as a side information, we also derive a finite-time logarithmic Bayes regret upper bound for a frequentist UCB algorithm. The bound changes only little as the prior becomes more informative, while the regret bound for the Bayesian algorithm eventually goes to zero. The bounds match asymptotically when $n \to \infty$ and the prior is overtaken by data.

3. To show the generality of our approach, we prove a $O(d\,c_\Delta \log^2 n)$ Bayes regret bound for a Bayesian linear bandit algorithm, where $d$ denotes the number of dimensions and $c_\Delta$ is a constant depending on random gaps. This bound also improves with a better prior.

4. Our analyses are a major departure from all recent Bayesian bandit analyses, starting with Russo and Van Roy [2014]. Roughly speaking, we first bound the regret in a fixed bandit instance, similarly to frequentist analyses, and then integrate out the random gap.

5. We show the tightness of our bounds empirically and compare them to prior bounds.

This paper is organized as follows. In Section 2, we introduce the setting of Bayesian bandits. In Section 3, we present a Bayesian upper confidence bound algorithm called BayesUCB [Kaufmann et al., 2012]. In Section 4, we derive finite-time logarithmic Bayes regret bounds for BayesUCB, in both multi-armed and linear bandits. These are the first such bounds ever derived. In Section 5, we compare our bounds to prior works and show that one matches an existing lower bound [Lai, 1987] asymptotically. In Section 6, we evaluate the bounds empirically. We conclude in Section 7.

## 2   Setting

We start with introducing our notation. Random variables are capitalized, except for Greek letters like $\theta$. For any positive integer $n$, we define $[n] = \{1, \ldots, n\}$. The indicator function is $\mathbb{1}\{\cdot\}$. The $i$-th entry of vector $v$ is $v_i$. If the vector is already indexed, such as $v_j$, we write $v_{j,i}$. We denote the maximum and minimum eigenvalues of matrix $M \in \mathbb{R}^{d \times d}$ by $\lambda_1(M)$ and $\lambda_d(M)$, respectively.

Our setting is defined as follows. We have a *multi-armed bandit* [Lai and Robbins, 1985, Lai, 1987, Auer et al., 2002, Abbasi-Yadkori et al., 2011] with an *action set* $\mathcal{A}$. Each *action* $a \in \mathcal{A}$ is associated with a *reward distribution* $p_a(\cdot; \theta)$, which is parameterized by an unknown *model parameter* $\theta$ shared by all actions. The learner interacts with the bandit instance for $n$ rounds indexed by $t \in [n]$. In each round $t$, it takes an *action* $A_t \in \mathcal{A}$ and observes its *stochastic reward* $Y_t \sim p_{A_t}(\cdot; \theta)$. The rewards are sampled independently across the rounds. We denote the mean of $p_a(\cdot; \theta)$ by $\mu_a(\theta)$ and call it

---

*The work started at Amazon Search.

---
**Algorithm 1** `BayesUCB`
---
1: **for** $t = 1, \dots, n$ **do**
2:      Compute the posterior distribution of $\theta$ using prior $h$ and history $H_t$
3:      **for** each action $a \in \mathcal{A}$ **do**
4:          Compute $U_{t,a}$ according to (1)
5:      Take action $A_t \leftarrow \arg\max_{a \in \mathcal{A}} U_{t,a}$ and observe its reward $Y_t$
---

the *mean reward* of action $a$. The optimal action is $A_* = \arg\max_{a \in \mathcal{A}} \mu_a(\theta)$ and its mean reward is $\mu_*(\theta) = \mu_{A_*}(\theta)$. For a fixed model parameter $\theta$, the $n$-round *regret* of a policy is defined as

$$R(n; \theta) = \mathbb{E}\left[ \sum_{t=1}^{n} \mu_*(\theta) - \mu_{A_t}(\theta) \,\middle|\, \theta \right],$$

where the expectation is taken over both random observations $Y_t$ and actions $A_t$. The *suboptimality gap* of action $a$ is $\Delta_a = \mu_*(\theta) - \mu_a(\theta)$ and the *minimum gap* is $\Delta_{\min} = \min_{a \in \mathcal{A} \setminus \{A_*\}} \Delta_a$.

Two settings are common in stochastic bandits. In the *frequentist* setting [Lai and Robbins, 1985, Auer et al., 2002, Abbasi-Yadkori et al., 2011], the learner has no additional information about $\theta$ and its objective is to minimize the worst-case regret for any bounded $\theta$. We study the *Bayesian* setting [Gittins, 1979, Lai, 1987, Russo and Van Roy, 2014, Russo et al., 2018], where the model parameter $\theta$ is drawn from a *prior distribution* $h$ that is given to the learner as a side information. The goal of the learner is to minimize the $n$-round *Bayes regret* $R(n) = \mathbb{E}\left[ R(n; \theta) \right]$, where the expectation is taken over the random model parameter $\theta \sim h$. Note that $A_*$, $\Delta_a$, and $\Delta_{\min}$ are random because they depend on the random instance $\theta$.

## 3 Algorithm

We study a Bayesian upper confidence bound algorithm called `BayesUCB` [Kaufmann et al., 2012]. The algorithm was analyzed in the Bayesian setting by Russo and Van Roy [2014]. The key idea in `BayesUCB` is to take the action with the highest UCB with respect to the posterior distribution of model parameter $\theta$. This differentiates it from frequentist algorithms, such as `UCB1` [Auer et al., 2002] and `LinUCB` [Abbasi-Yadkori et al., 2011], where the UCBs are computed using a frequentist *maximum likelihood estimate (MLE)* of the model parameter.

Let $H_t = (A_\ell, Y_\ell)_{\ell \in [t-1]}$ be the *history* of taken actions and their observed rewards up to round $t$. The *Bayesian UCB* for the mean reward of action $a$ at round $t$ is

$$U_{t,a} = \mu_a(\hat{\theta}_t) + C_{t,a}, \tag{1}$$

where $\hat{\theta}_t$ is the *posterior mean estimate* of $\theta$ at round $t$ and $C_{t,a}$ is a *confidence interval width* for action $a$ at round $t$. The posterior distribution of model parameter $\theta$ is computed from a prior $h$ and history $H_t$ using Bayes' rule. The width is chosen so that $|\mu_a(\hat{\theta}_t) - \mu_a(\theta)| \le C_{t,a}$ holds with a high probability conditioned on any history $H_t$. Technically speaking, $C_{t,a}$ is a half-width but we call it a width to simplify terminology.

Our algorithm is presented in Algorithm 1. We instantiate it in a Gaussian bandit in Section 3.1, in a Bernoulli bandit in Section 3.2, and in a linear bandit with Gaussian rewards in Section 3.3. These settings are of practical interest because they lead to computationally-efficient implementations that can be analyzed due to closed-form posteriors [Lu and Van Roy, 2019, Kveton et al., 2021, Basu et al., 2021, Wang et al., 2021, Hong et al., 2022]. While we focus on deriving logarithmic Bayes regret bounds for `BayesUCB`, we believe that similar analyses can be done for Thompson sampling [Thompson, 1933, Chapelle and Li, 2012, Agrawal and Goyal, 2012, 2013, Russo and Van Roy, 2014, Russo et al., 2018]. This extension is non-trivial because a key step in our analysis is that the action with the highest UCB is taken (Section 5.3).

### 3.1 Gaussian Bandit

In a $K$-armed Gaussian bandit, the action set is $\mathcal{A} = [K]$ and the model parameter is $\theta \in \mathbb{R}^K$. Each action $a \in \mathcal{A}$ has a Gaussian reward distribution, $p_a(\cdot; \theta) = \mathcal{N}(\cdot; \theta_a, \sigma^2)$, where $\theta_a$ is its mean and

$\sigma > 0$ is a known reward noise. Thus $\mu_a(\theta) = \theta_a$. The model parameter $\theta$ is drawn from a known Gaussian prior $h(\cdot) = \mathcal{N}(\cdot; \mu_0, \sigma_0^2 I_K)$, where $\mu_0 \in \mathbb{R}^K$ is a vector of prior means and $\sigma_0 > 0$ is a prior width.

The posterior distribution of the mean reward of action $a$ at round $t$ is $\mathcal{N}(\cdot; \hat{\theta}_{t,a}, \hat{\sigma}_{t,a}^2)$, where

$$\hat{\sigma}_{t,a}^2 = (\sigma_0^{-2} + \sigma^{-2} N_{t,a})^{-1}$$

is the posterior variance, $N_{t,a} = \sum_{\ell=1}^{t-1} \mathbb{1}\{A_\ell = a\}$ is the number of observations of action $a$ up to round $t$, and

$$\hat{\theta}_{t,a} = \hat{\sigma}_{t,a}^2 \left( \sigma_0^{-2} \mu_{0,a} + \sigma^{-2} \sum_{\ell=1}^{t-1} \mathbb{1}\{A_\ell = a\} Y_\ell \right)$$

is the posterior mean. This follows from a classic result, that the posterior distribution of the mean of a Gaussian random variable with a Gaussian prior is a Gaussian [Bishop, 2006]. The Bayesian UCB of action $a$ at round $t$ is $U_{t,a} = \hat{\theta}_{t,a} + C_{t,a}$, where $C_{t,a} = \sqrt{2\hat{\sigma}_{t,a}^2 \log(1/\delta)}$ is the confidence interval width and $\delta \in (0,1)$ is a failure probability of the confidence interval.

### 3.2 Bernoulli Bandit

In a $K$-armed Bernoulli bandit, the action set is $\mathcal{A} = [K]$ and the model parameter is $\theta \in \mathbb{R}^K$. Each action $a \in \mathcal{A}$ has a Bernoulli reward distribution, $p_a(\cdot; \theta) = \mathrm{Ber}(\cdot; \theta_a)$, where $\theta_a$ is its mean. Hence $\mu_a(\theta) = \theta_a$. Each parameter $\theta_a$ is drawn from a known prior $\mathrm{Beta}(\cdot; \alpha_a, \beta_a)$, where $\alpha_a > 0$ and $\beta_a > 0$ are positive and negative prior pseudo-counts, respectively.

The posterior distribution of the mean reward of action $a$ at round $t$ is $\mathrm{Beta}(\cdot; \alpha_{t,a}, \beta_{t,a})$, where

$$\alpha_{t,a} = \alpha_a + \sum_{\ell=1}^{t-1} \mathbb{1}\{A_\ell = a\} Y_\ell, \quad \beta_{t,a} = \beta_a + \sum_{\ell=1}^{t-1} \mathbb{1}\{A_\ell = a\} (1 - Y_\ell).$$

This follows from a classic result, that the posterior distribution of the mean of a Bernoulli random variable with a beta prior is a beta distribution [Bishop, 2006]. The corresponding Bayesian UCB is $U_{t,a} = \hat{\theta}_{t,a} + C_{t,a}$, where

$$\hat{\theta}_{t,a} = \frac{\alpha_{t,a}}{\alpha_{t,a} + \beta_{t,a}}, \quad C_{t,a} = \sqrt{\frac{\log(1/\delta)}{2(\alpha_{t,a} + \beta_{t,a} + 1)}} = \sqrt{\frac{\log(1/\delta)}{2(\alpha_a + \beta_a + N_{t,a} + 1)}},$$

denote the posterior mean and confidence interval width, respectively, of action $a$ at round $t$; and $\delta \in (0,1)$ is a failure probability of the confidence interval. The confidence interval is derived using the fact that $\mathrm{Beta}(\cdot; \alpha_{t,a}, \beta_{t,a})$ is a sub-Gaussian distribution with variance proxy $\frac{1}{4(\alpha_a + \beta_a + N_{t,a} + 1)}$ [Marchal and Arbel, 2017].

### 3.3 Linear Bandit with Gaussian Rewards

We also study linear bandits [Dani et al., 2008, Abbasi-Yadkori et al., 2011] with a finite number of actions $\mathcal{A} \subseteq \mathbb{R}^d$ in $d$ dimensions. The model parameter is $\theta \in \mathbb{R}^d$. All actions $a \in \mathcal{A}$ have Gaussian reward distributions, $p_a(\cdot; \theta) = \mathcal{N}(\cdot; a^\top \theta, \sigma^2)$, where $\sigma > 0$ is a known reward noise. Therefore, the mean reward of action $a$ is $\mu_a(\theta) = a^\top \theta$. The parameter $\theta$ is drawn from a known multivariate Gaussian prior $h(\cdot) = \mathcal{N}(\cdot; \theta_0, \Sigma_0)$, where $\theta_0 \in \mathbb{R}^d$ is its mean and $\Sigma_0 \in \mathbb{R}^{d \times d}$ is its covariance, represented by a *positive semi-definite (PSD)* matrix.

The posterior distribution of $\theta$ at round $t$ is $\mathcal{N}(\cdot; \hat{\theta}_t, \hat{\Sigma}_t)$, where

$$\hat{\theta}_t = \hat{\Sigma}_t \left( \Sigma_0^{-1} \theta_0 + \sigma^{-2} \sum_{\ell=1}^{t-1} A_\ell Y_\ell \right), \quad \hat{\Sigma}_t = (\Sigma_0^{-1} + G_t)^{-1}, \quad G_t = \sigma^{-2} \sum_{\ell=1}^{t-1} A_\ell A_\ell^\top.$$

Here $\hat{\theta}_t$ and $\hat{\Sigma}_t$ are the posterior mean and covariance of $\theta$, respectively, and $G_t$ is the outer product of the feature vectors of the taken actions up to round $t$. These formulas follow from a classic result,

that the posterior distribution of a linear model parameter with a Gaussian prior and observations is a Gaussian [Bishop, 2006]. The Bayesian UCB of action $a$ at round $t$ is $U_{t,a} = a^\top \hat{\theta}_t + C_{t,a}$, where $C_{t,a} = \sqrt{2 \log(1/\delta)} \|a\|_{\hat{\Sigma}_t}$ is the confidence interval width, $\delta \in (0,1)$ is a failure probability of the confidence interval, and $\|a\|_M = \sqrt{a^\top M a}$.

# 4  Logarithmic Bayes Regret Upper Bounds

In this section, we present finite-time logarithmic Bayes regret bounds for `BayesUCB`. We derive them for both $K$-armed and linear bandits. One bound matches an existing lower bound of Lai [1987] asymptotically and all improve upon prior $\tilde{O}(\sqrt{n})$ bounds. We discuss this in detail in Section 5.

## 4.1  `BayesUCB` in Gaussian Bandit

Our first regret bound is for `BayesUCB` in a $K$-armed Gaussian bandit. It depends on random gaps. To control the gaps, we clip them as $\Delta_a^\varepsilon = \max \{\Delta_a, \varepsilon\}$. The bound is stated below.

**Theorem 1.** *For any $\varepsilon > 0$ and $\delta \in (0,1)$, the $n$-round Bayes regret of `BayesUCB` in a $K$-armed Gaussian bandit is bounded as*

$$R(n) \leq \mathbb{E} \left[ \sum_{a \neq A_*} \frac{8\sigma^2 \log(1/\delta)}{\Delta_a^\varepsilon} - \frac{\sigma^2 \Delta_a^\varepsilon}{\sigma_0^2} \right] + C \,,$$

*where $C = \varepsilon n + 2(\sqrt{2 \log(1/\delta)} + 2K)\sigma_0 K n \delta$ is a low-order term.*

The proof is in Appendix A.1. For $\varepsilon = 1/n$ and $\delta = 1/n$, the bound is $O(c_\Delta \log n)$, where $c_\Delta$ is a constant depending on the gaps of random bandit instances. The dependence on $\sigma_0$ in the low-order term $C$ can be reduced to $\min \{\sigma_0, \sigma\}$ by a more elaborate analysis, where the regret of taking each action for the first time is bounded separately. This also applies to Corollary 2.

Now we derive an upper bound on Theorem 1 that eliminates the dependence on random gaps. To state it, we need to introduce additional notation. For any action $a$, we denote all action parameters except for $a$ by $\theta_{-a} = (\theta_1, \ldots, \theta_{a-1}, \theta_{a+1}, \ldots, \theta_K)$ and the corresponding optimal action in $\theta_{-a}$ by $\theta_a^* = \max_{j \in \mathcal{A} \setminus \{a\}} \theta_j$. We denote by $h_a$ the prior density of $\theta_a$ and by $h_{-a}$ the prior density of $\theta_{-a}$. Since the prior is factored (Section 3.1), note that $h(\theta) = h_a(\theta_a) h_{-a}(\theta_{-a})$ for any $\theta$ and action $a$. To keep the result clean, we state it for a "sufficiently" large prior variance. A complete statement for all prior variances is given in Appendix B. We note that the setting of small prior variances favors Bayesian algorithms since their regret decreases with a more informative prior. In fact, we show in Appendix B that the regret of `BayesUCB` is $O(1)$ for a sufficiently small $\sigma_0$.

**Corollary 2.** *Let $\sigma_0^2 \geq \frac{1}{8 \log(1/\delta) \, n^2 \log \log n}$. Then there exist functions $\xi_a : \mathbb{R} \to \left[ \frac{1}{n}, \frac{1}{\sqrt{\log n}} \right]$ such that the $n$-round Bayes regret of `BayesUCB` in a $K$-armed Gaussian bandit is bounded as*

$$R(n) \leq \left[ 8\sigma^2 \log(1/\delta) \log n - \frac{\sigma^2}{2\sigma_0^2 \log n} \right] \sum_{a \in \mathcal{A}} \int_{\theta_{-a}} h_a(\theta_a^* - \xi_a(\theta_a^*)) \, h_{-a}(\theta_{-a}) \, \mathrm{d}\theta_{-a} + C \,,$$

*where $C = 8\sigma^2 K \log(1/\delta)\sqrt{\log n} + 2(\sqrt{2 \log(1/\delta)} + 2K)\sigma_0 K n \delta + 1$ is a low-order term.*

The proof is in Appendix A.2. For $\delta = 1/n$, the bound is $O(c_h \log^2 n)$, where $c_h$ depends on prior $h$ but not on the gaps of random bandit instances. This bound is motivated by Lai [1987]. The terms $\xi_a$ arise due to the intermediate value theorem for function $h_a$. Similar terms appear in Lai [1987] but vanish in their final asymptotic claims. The rate $1/\sqrt{\log n}$ in the definition of $\xi_a$ cannot be reduced to $1/\text{polylog } n$ without increasing dependence on $n$ in other parts of the bound.

The complexity term $\sum_{a \in \mathcal{A}} \int_{\theta_{-a}} h_a(\theta_a^* - \xi_a(\theta_a^*)) \, h_{-a}(\theta_{-a}) \, \mathrm{d}\theta_{-a}$ in Corollary 2 is the same as in Lai [1987] and can be interpreted as follows. Consider the asymptotic regime of $n \to \infty$. Then, since the range of $\xi_a$ is $\left[ \frac{1}{n}, \frac{1}{\sqrt{\log n}} \right]$, the term simplifies to $\sum_{a \in \mathcal{A}} \int_{\theta_{-a}} h_a(\theta_a^*) h_{-a}(\theta_{-a}) \, \mathrm{d}\theta_{-a}$ and can be viewed as the distance between prior means. In a Gaussian bandit with $K = 2$ actions, it has a closed form of $\frac{1}{\sqrt{\pi \sigma_0^2}} \exp \left[ -\frac{(\mu_{0,1} - \mu_{0,2})^2}{4\sigma_0^2} \right]$. A general upper bound for $K > 2$ actions is given below.

**Lemma 3.** *In a $K$-armed Gaussian bandit with prior $h(\cdot) = \mathcal{N}(\cdot; \mu_0, \sigma_0^2 I_K)$, we have*

$$\sum_{a \in \mathcal{A}} \int_{\theta_{-a}} h_a(\theta_a^*) \, h_{-a}(\theta_{-a}) \, \mathrm{d}\theta_{-a} \leq \frac{1}{2\sqrt{\pi\sigma_0^2}} \sum_{a \in \mathcal{A}} \sum_{a' \neq a} \exp\left[-\frac{(\mu_{0,a} - \mu_{0,a'})^2}{4\sigma_0^2}\right].$$

The bound is proved in Appendix A.3 and has several interesting properties that capture low-regret regimes. First, as the prior becomes more informative and concentrated, $\sigma_0 \to 0$, the bound goes to zero. Second, when the gaps of bandit instances sampled from the prior are large, low regret is also expected. This can happen when the prior means become more separated, $|\mu_{0,a} - \mu_{0,a'}| \to \infty$, or the prior becomes wider, $\sigma_0 \to \infty$. Our bound goes to zero in both of these cases. This also implies that Bayes regret bounds are not necessarily monotone in prior parameters, such as $\sigma_0$.

## 4.2  `UCB1` in Gaussian Bandit

Using a similar approach, we prove a Bayes regret bound for `UCB1` [Auer et al., 2002]. We view it as `BayesUCB` (Section 3.1) where $\sigma_0 = \infty$ and each action $a \in \mathcal{A}$ is initially taken once at round $t = a$. This generalizes classic `UCB1` to $\sigma^2$-sub-Gaussian noise. An asymptotic Bayes regret bound for `UCB1` was proved by Lai [1987] (claim (i) in their Theorem 3). We derive a finite-time prior-dependent Bayes regret bound below.

**Theorem 4.** *There exist functions $\xi_a : \mathbb{R} \to \left[\frac{1}{n}, \frac{1}{\sqrt{\log n}}\right]$ such that the $n$-round Bayes regret of `UCB1` in a $K$-armed Gaussian bandit is bounded as*

$$R(n) \leq 8\sigma^2 \log(1/\delta) \log n \sum_{a \in \mathcal{A}} \int_{\theta_{-a}} h_a(\theta_a^* - \xi_a(\theta_a^*)) \, h_{-a}(\theta_{-a}) \, \mathrm{d}\theta_{-a} + C,$$

*where $C = 8\sigma^2 K \log(1/\delta)\sqrt{\log n} + 2(\sqrt{2\log(1/\delta)} + 2K)\sigma Kn\delta + \sum_{a \in \mathcal{A}} \mathbb{E}[\Delta_a] + 1$.*

The proof is in Appendix A.4. For $\delta = 1/n$, the bound is $O(c_h \log^2 n)$ and similar to Corollary 2. The main difference is in the additional factor $\frac{\sigma^2}{2\sigma_0^2 \log n}$ in Corollary 2, which decreases the bound. This means that the regret of `BayesUCB` improves as $\sigma_0$ decreases while that of `UCB1` may not change much. In fact, the regret bound of `BayesUCB` is $O(1)$ as $\sigma_0 \to 0$ (Appendix B) while that of `UCB1` remains logarithmic. This is expected because `BayesUCB` has more information about the random instance $\theta$ as $\sigma_0$ decreases, while the frequentist algorithm is oblivious to the prior.

## 4.3  `BayesUCB` in Bernoulli Bandit

Theorem 1 and Corollary 2 can be straightforwardly extended to Bernoulli bandits because

$$\mathbb{P}\left(|\theta_a - \hat{\theta}_{t,a}| \geq C_{t,a} \,\middle|\, H_t\right) \leq 2\delta$$

holds for any action $a$ and history $H_t$ (Section 3.2). We state the extension below and prove it in Appendix A.5.

**Theorem 5.** *For any $\varepsilon > 0$ and $\delta \in (0, 1)$, the $n$-round Bayes regret of `BayesUCB` in a $K$-armed Bernoulli bandit is bounded as*

$$R(n) \leq \mathbb{E}\left[\sum_{a \neq A_*} \frac{2\log(1/\delta)}{\Delta_a^\varepsilon} - (\alpha_a + \beta_a + 1)\Delta_a^\varepsilon\right] + C,$$

*where $C = \varepsilon n + 2Kn\delta$ is a low-order term.*

*Moreover, let $\lambda = \min_{a \in \mathcal{A}} \alpha_a + \beta_a + 1$ and $\lambda \leq 2\log(1/\delta)\, n^2 \log\log n$. Then*

$$R(n) \leq \left[2\log(1/\delta)\log n - \frac{\lambda}{2\log n}\right] \sum_{a \in \mathcal{A}} \int_{\theta_{-a}} h_a(\theta_a^* - \xi_a(\theta_a^*)) \, h_{-a}(\theta_{-a}) \, \mathrm{d}\theta_{-a} + C,$$

*where $C = 2K\log(1/\delta)\sqrt{\log n} + 2Kn\delta + 1$ is a low-order term.*

### 4.4 `BayesUCB` in Linear Bandit

Now we present a gap-dependent Bayes regret bound for `BayesUCB` in a linear bandit with a finite number of actions. The bound depends on a random minimum gap. To control the gap, we clip it as $\Delta_{\min}^{\varepsilon} = \max\{\Delta_{\min}, \varepsilon\}$.

**Theorem 6.** *Suppose that $\|\theta\|_2 \leq L_*$ holds with probability at least $1 - \delta_*$. Let $\|a\|_2 \leq L$ for any action $a \in \mathcal{A}$. Then for any $\varepsilon > 0$ and $\delta \in (0,1)$, the $n$-round Bayes regret of linear `BayesUCB` is bounded as*

$$
R(n) \leq 8 \mathbb{E}\left[\frac{1}{\Delta_{\min}^{\varepsilon}}\right] \frac{\sigma_{0,\max}^2 d}{\log\left(1 + \frac{\sigma_{0,\max}^2}{\sigma^2}\right)} \log\left(1 + \frac{\sigma_{0,\max}^2 n}{\sigma^2 d}\right) \log(1/\delta) + \varepsilon n + 4 L L_* K n \delta
$$

*with probability at least $1 - \delta_*$, where $\sigma_{0,\max} = \sqrt{\lambda_1(\Sigma_0)} L$.*

The proof is in Appendix A.6. For $\varepsilon = 1/n$ and $\delta = 1/n$, the bound is $O(d\, c_\Delta \log^2 n)$, where $c_\Delta$ is a constant depending on the gaps of random bandit instances. The bound is remarkably similar to the frequentist $O(d\, \Delta_{\min}^{-1} \log^2 n)$ bound in Theorem 5 of Abbasi-Yadkori et al. [2011], where $\Delta_{\min}$ is the minimum gap. There are two differences. First, we integrate $\Delta_{\min}^{-1}$ over the prior. Second, our bound decreases as the prior becomes more informative, $\sigma_{0,\max} \to 0$.

In a Gaussian bandit, the bound becomes $O(K \mathbb{E}\left[1/\Delta_{\min}^{\varepsilon}\right] \log^2 n)$. Therefore, it is comparable to Theorem 1 up to an additional logarithmic factor in $n$. This is due to a more general proof technique, which allows for dependencies between the mean rewards of actions. We also note that Theorem 6 does not assume that the prior is factored, unlike Theorem 1.

## 5 Comparison to Prior Works

This section is organized as follows. In Section 5.1, we show that the bound in Theorem 5 matches an existing lower bound of Lai [1987] asymptotically. In Section 5.2, we compare our logarithmic bounds to prior $\tilde{O}(\sqrt{n})$ bounds. Finally, in Section 5.3, we outline the key steps in our analyses and how they differ from prior works.

### 5.1 Matching Lower Bound

In frequentist bandit analyses, it is standard to compare asymptotic lower bounds to finite-time upper bounds because finite-time logarithmic lower bounds do not exist [Lattimore and Szepesvari, 2019]. We follow the same approach when arguing that our finite-time upper bounds are order optimal.

The results in Lai [1987] are for single-parameter exponential-family reward distributions, which excludes Gaussian rewards. Therefore, we argue about the tightness of Theorem 5 only. Specifically, we take the second bound in Theorem 5, set $\delta = 1/n$, and let $n \to \infty$. In this case, $\frac{\lambda}{2 \log n} \to 0$ and $\xi_a(\cdot) \to 0$, and the bound matches up to constant factors the lower bound in Lai [1987] (claim (ii) in their Theorem 3), which is

$$
\Omega\left(\log^2 n \sum_{a \in \mathcal{A}} \int_{\theta_{-a}} h_a(\theta_a^*)\, h_{-a}(\theta_{-a})\, \mathrm{d}\theta_{-a}\right). \tag{2}
$$

Lai [1987] also matched this lower bound with an asymptotic upper bound for a frequentist policy.

Our finite-time upper bounds also reveal an interesting difference from the asymptotic lower bound in (2), which may deserve more future attention. More specifically, the regret bound of `BayesUCB` (Corollary 2) improves with prior information while that of `UCB1` (Theorem 4) does not. We observe these improvements empirically as well (Section 6 and Appendix D). However, both bounds are the same asymptotically. This is because the benefit of knowing the prior vanishes in asymptotic analyses, since $\frac{\sigma^2}{2\sigma_0^2 \log n} \to 0$ in Corollary 2 as $n \to \infty$. This motivates the need for finite-time logarithmic Bayes regret lower bounds, which do not exist.

## 5.2 Prior Bayes Regret Upper Bounds

Theorem 1 and Corollary 2 are major improvements upon existing $\tilde{O}(\sqrt{n})$ bounds. For instance, take a prior-dependent bound in Lemma 4 of Kveton et al. [2021], which holds for both `BayesUCB` and Thompson sampling due to a well-known equivalence of their analyses [Russo and Van Roy, 2014, Hong et al., 2020]. For $\delta = 1/n$, their leading term becomes

$$4\sqrt{2\sigma^2 K \log n}\left(\sqrt{n + \sigma^2 \sigma_0^{-2} K} - \sqrt{\sigma^2 \sigma_0^{-2} K}\right). \tag{3}$$

Similarly to Theorem 1 and Corollary 2, (3) decreases as the prior concentrates and becomes more informative, $\sigma_0 \to 0$. However, the bound is $\tilde{O}(\sqrt{n})$. Moreover, it does not depend on prior means $\mu_0$ or the gaps of random bandit instances. Therefore, it cannot capture low-regret regimes due to large random gaps $\Delta_a^\varepsilon$ in Theorem 1 or a small complexity term in Corollary 2. We demonstrate it empirically in Section 6.

When the random gaps $\Delta_a^\varepsilon$ in Theorem 1 are small or the complexity term in Corollary 2 is large, our bounds can be worse than $\tilde{O}(\sqrt{n})$ bounds. This is analogous to the relation of the gap-dependent and gap-free frequentist bounds [Lattimore and Szepesvari, 2019]. Specifically, a gap-dependent bound of `UCB1` in a $K$-armed bandit with 1-sub-Gaussian rewards (Theorem 7.1) is $O(K\Delta_{\min}^{-1}\log n)$, where $\Delta_{\min}$ is the minimum gap. A corresponding gap-free bound (Theorem 7.2) is $O(\sqrt{Kn\log n})$. The latter is smaller when the gap is small, $\Delta_{\min} = o(\sqrt{(K\log n)/n})$. To get the best bound, the minimum of the two should be taken, and the same is true in our Bayesian setting.

No prior-dependent Bayes regret lower bound exists in linear bandits. Thus we treat $\mathbb{E}\left[1/\Delta_{\min}^\varepsilon\right]$ in Theorem 6 as the complexity term and do not further bound it as in Corollary 2. To compare our bound fairly to existing $\tilde{O}(\sqrt{n})$ bounds, we derive an $\tilde{O}(\sqrt{n})$ bound in Appendix C, by a relatively minor change in the proof of Theorem 6. A similar bound can be obtained by adapting the proofs of Lu and Van Roy [2019] and Hong et al. [2022] to a linear bandit with a finite number of actions. The leading term of the bound is

$$2\sqrt{\frac{2\sigma_{0,\max}^2 dn}{\log\left(1 + \frac{\sigma_{0,\max}^2}{\sigma^2}\right)}\log\left(1 + \frac{\sigma_{0,\max}^2 n}{\sigma^2 d}\right)\log(1/\delta)}. \tag{4}$$

Similarly to Theorem 6, (4) decreases as the prior becomes more informative, $\sigma_{0,\max} \to 0$. However, the bound is $\tilde{O}(\sqrt{n})$ and does not depend on the gaps of random bandit instances. Hence it cannot capture low-regret regimes due to a large random minimum gap $\Delta_{\min}$ in Theorem 6. We validate it empirically in Section 6.

## 5.3 Technical Novelty

All modern Bayesian analyses follow Russo and Van Roy [2014], who derived the first finite-time $\tilde{O}(\sqrt{n})$ Bayes regret bounds for `BayesUCB` and Thompson sampling. The key idea in their analyses is that conditioned on history, the optimal and taken actions are identically distributed, and that the upper confidence bounds are deterministic functions of the history. This is where the randomness of instances in Bayesian bandits is used. Using this, the regret at round $t$ is bounded by the confidence interval width of the taken action, and the usual $\tilde{O}(\sqrt{n})$ bounds can be obtained by summing up the confidence interval widths over $n$ rounds.

The main difference in our work is that we first bound the regret in a fixed bandit instance, similarly to frequentist analyses. The bound involves $\Delta_a^{-1}$ and is derived using biased Bayesian confidence intervals. The rest of our analysis is Bayesian in two parts: we prove that the confidence intervals fail with a low probability and bound random $\Delta_a^{-1}$, following a similar technique to Lai [1987]. The resulting logarithmic Bayes regret bounds cannot be derived using the techniques of Russo and Van Roy [2014], as these become loose when the confidence interval widths are introduced.

Asymptotic logarithmic Bayes regret bounds were derived in Lai [1987]. From this analysis, we use only the technique for bounding $\Delta_a^{-1}$ when proving Corollary 2 and Theorem 5. The central part of our proof is a finite-time per-instance bound on the number of times that a suboptimal action is taken. This quantity is bounded based on the assumption that the action with the highest UCB is taken. A

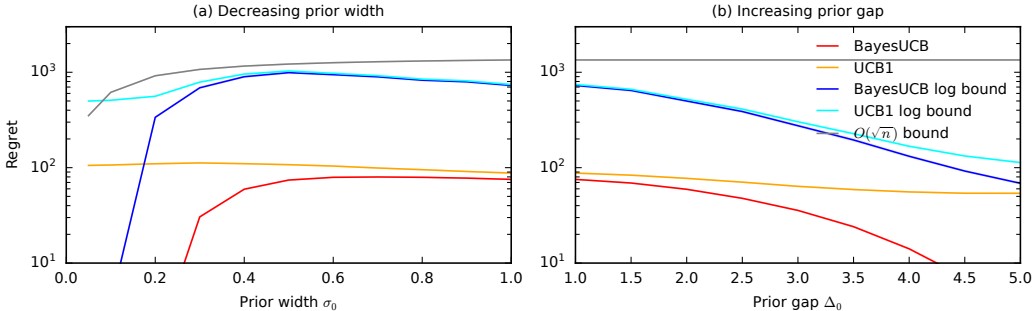

Figure 1: Gaussian bandit as (a) the prior width $\sigma_0$ and (b) the prior gap $\Delta_0$ change.

comparable argument in Theorem 2 of Lai [1987] is asymptotic and on average over random bandit instances.

# 6 Experiments

We experiment with UCB algorithms in two environments: Gaussian bandits (Section 3.1) and linear bandits with Gaussian rewards (Section 3.3). In both experiments, the horizon is $n = 1\,000$ rounds. All results are averaged over $10\,000$ random runs. Shaded regions in the plots are standard errors of the estimates. They are generally small because the number of runs is high.

## 6.1 Gaussian Bandit

The first problem is a $K$-armed bandit with $K = 10$ actions (Section 3.1). The *prior width* is $\sigma_0 = 1$. The prior mean is $\mu_0$, where $\mu_{0,1} = \Delta_0$ and $\mu_{0,a} = 0$ for $a > 1$. We set $\Delta_0 = 1$ and call it the *prior gap*. We vary $\sigma_0$ and $\Delta_0$ in our experiments, and observe how the regret and its upper bounds change as the problem hardness varies (Sections 4.1 and 5.2).

We plot five trends: (a) Bayes regret of BayesUCB. (b) Bayes regret of UCB1 (Section 4.2), which is a comparable frequentist algorithm to BayesUCB. (c) A leading term of the BayesUCB regret bound in Theorem 1, where $\varepsilon = 1/n$ and $\delta = 1/n$. (d) A leading term of the UCB1 regret bound: This is the same as (c) with $\sigma_0 = \infty$. (e) An existing $\tilde{O}(\sqrt{n})$ regret bound in (3).

Our results are reported in Figure 1. We observe three major trends. First, the regret of BayesUCB decreases as the problem becomes easier, either $\sigma_0 \to 0$ or $\Delta_0 \to \infty$. It is also lower than that of UCB1, which does not leverage the prior. Second, the regret bound of BayesUCB is tighter than that of UCB1, due to capturing the benefit of the prior. Finally, the logarithmic regret bounds are tighter than the $\tilde{O}(\sqrt{n})$ bound. In addition, the $\tilde{O}(\sqrt{n})$ bound depends on the prior only through $\sigma_0$ and thus remains constant as the prior gap $\Delta_0$ changes.

In Appendix D, we compare BayesUCB to UCB1 more comprehensively for various $K$, $\sigma$, $\Delta_0$, and $\sigma_0$. In all experiments, BayesUCB has a lower regret than UCB1. This also happens when the noise is not Gaussian, which a testament to the robustness of Bayesian methods to model misspecification.

## 6.2 Linear Bandit with Gaussian Rewards

The second problem is a linear bandit with $K = 30$ actions in $d = 10$ dimensions (Section 3.3). The prior covariance is $\Sigma_0 = \sigma_0^2 I_d$. The prior mean is $\theta_0$, where $\theta_{0,1} = \Delta_0$ and $\theta_{0,i} = -1$ for $i > 1$. As in Section 6.1, we set $\Delta_0 = 1$ and call it the *prior gap*. The action set $\mathcal{A}$ is generated as follows. The first $d$ actions are the canonical basis in $\mathbb{R}^d$. The remaining $K - d$ actions are sampled uniformly at random from the positive orthant and scaled to unit length. This ensures that the first action has the highest mean reward, of $\Delta_0$, under the prior mean $\theta_0$. We vary $\sigma_0$ and $\Delta_0$, and observe how the regret and its upper bounds change as the problem hardness varies (Sections 4.4 and 5.2). We plot three trends: (a) Bayes regret of BayesUCB. (b) A leading term of the BayesUCB regret bound in Theorem 6, where $\sigma_{0,\max} = \sigma_0$, $\varepsilon = 1/n$, and $\delta = 1/n$. (c) An existing $\tilde{O}(\sqrt{n})$ regret bound in (4).

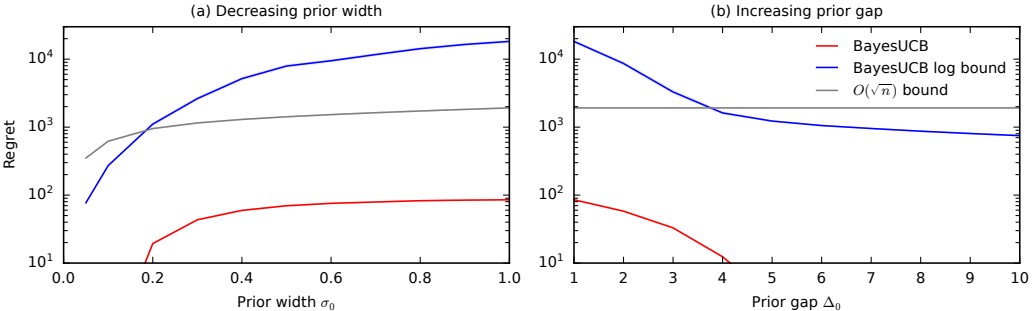

Figure 2: Linear bandit as (a) the prior width $\sigma_0$ and (b) the prior gap $\Delta_0$ change.

Our results are reported in Figure 2 and we observe three major trends. First, the regret of `BayesUCB` decreases as the problem becomes easier, either $\sigma_0 \to 0$ or $\Delta_0 \to \infty$. Second, the regret bound of `BayesUCB` decreases as the problem becomes easier. Finally, our logarithmic regret bound can also be tighter than the $\tilde{O}(\sqrt{n})$ bound. In particular, the $\tilde{O}(\sqrt{n})$ bound depends on the prior only through $\sigma_0$, and thus remains constant as the prior gap $\Delta_0$ changes. We discuss when our bounds could be looser than $\tilde{O}(\sqrt{n})$ bounds in Section 5.2.

## 7 Conclusions

Finite-time logarithmic frequentist regret bounds are the standard way of analyzing $K$-armed bandits [Auer et al., 2002, Garivier and Cappe, 2011, Agrawal and Goyal, 2012]. In our work, we prove the first comparable finite-time bounds, logarithmic in $n$, in the Bayesian setting. This is a major step in theory and a significant improvement upon prior $\tilde{O}(\sqrt{n})$ Bayes regret bounds that have become standard in the literature. Comparing to frequentist regret bounds, our bounds capture the value of prior information given to the learner. Our proof technique is general and we also apply it to linear bandits.

This work can be extended in many directions. First, our analyses only need closed-form posteriors, which are available for other reward distributions than Gaussian and Bernoulli. Second, our linear bandit analysis (Section 4.4) seems preliminary when compared to our multi-armed bandit analyses. As an example, the complexity term $\mathbb{E}\left[1/\Delta_{\min}^{\varepsilon}\right]$ in Theorem 6 could be bounded as in Corollary 2 and Theorem 5. We do not do this because the main reason for deriving the $O(c_h \log^2 n)$ bound in Theorem 5, an upper bound on the corresponding $O(c_\Delta \log n)$ bound, is that it matches the lower bound in (2). No such instance-dependent lower bound exists in Bayesian linear bandits. Third, we believe that our approach can be extended to information-theory bounds [Russo and Van Roy, 2016] and discuss it in Appendix E. Fourth, although we only analyze `BayesUCB`, we believe that similar guarantees can be obtained for Thompson sampling. Finally, we would like to extend our results to reinforcement learning, for instance by building on the work of Lu and Van Roy [2019].

There have been recent attempts in theory [Wagenmaker and Foster, 2023] to design general adaptive algorithms with finite-time instance-dependent bounds based on optimal allocations. The promise of these methods is a higher statistical efficiency than exploring by optimism, which we adopt in this work. One of their shortcomings is that they are not guaranteed to be computationally efficient, as discussed in Section 2.2 of Wagenmaker and Foster [2023]. This work is also frequentist.

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

# A Proofs

We present a general approach for deriving finite-time logarithmic Bayes regret bounds. We start with $K$-armed bandits and then extend it to linear bandits.

## A.1 Proof of Theorem 1

Let $E_t = \left\{ \forall a \in \mathcal{A} : |\theta_a - \hat{\theta}_{t,a}| \leq C_{t,a} \right\}$ be the event that all confidence intervals at round $t$ hold. Fix $\varepsilon > 0$. We start with decomposing the $n$-round regret as

$$\sum_{t=1}^n \mathbb{E}\left[\Delta_{A_t}\right] \leq \sum_{t=1}^n \mathbb{E}\left[\Delta_{A_t} \mathbb{1}\{\Delta_{A_t} \geq \varepsilon, E_t\}\right] + \sum_{t=1}^n \mathbb{E}\left[\Delta_{A_t} \mathbb{1}\{\Delta_{A_t} < \varepsilon\}\right] + \tag{5}$$
$$\sum_{t=1}^n \mathbb{E}\left[\Delta_{A_t} \mathbb{1}\{\bar{E}_t\}\right] .$$

We bound the first term using the design of `BayesUCB` and its closed-form posteriors.

**Case 1: Event $E_t$ occurs and the gap is large, $\Delta_{A_t} \geq \varepsilon$.** Since $E_t$ occurs and the action with the highest UCB is taken, the regret at round $t$ can be bounded as

$$\Delta_{A_t} = \theta_{A_*} - \theta_{A_t} \leq \theta_{A_*} - U_{t,A_*} + U_{t,A_t} - \theta_{A_t} \leq U_{t,A_t} - \theta_{A_t} \leq 2C_{t,A_t} .$$

In the second inequality, we use that $\theta_{A_*} \leq U_{t,A_*}$ on event $E_t$. This implies that on event $E_t$, action $a$ can be taken only if

$$\Delta_a \leq 2C_{t,a} = 2\sqrt{2\hat{\sigma}_{t,a}^2 \log(1/\delta)} = 2\sqrt{\frac{2\log(1/\delta)}{\sigma_0^{-2} + \sigma^{-2} N_{t,a}}} ,$$

which can be rearranged as

$$N_{t,a} \leq \frac{8\sigma^2 \log(1/\delta)}{\Delta_a^2} - \sigma^2 \sigma_0^{-2} .$$

Therefore, the number of times that action $a$ is taken in $n$ rounds while all confidence intervals hold, $N_a = \sum_{t=1}^n \mathbb{1}\{A_t = a, E_t\}$, is bounded as

$$N_a \leq \frac{8\sigma^2 \log(1/\delta)}{\Delta_a^2} - \sigma^2 \sigma_0^{-2} . \tag{6}$$

Now we apply this inequality to bound the first term in (5) as

$$\sum_{t=1}^n \mathbb{E}\left[\Delta_{A_t} \mathbb{1}\{\Delta_{A_t} \geq \varepsilon, E_t\}\right] \leq \mathbb{E}\left[\sum_{a \neq A_*} \left(\frac{8\sigma^2 \log(1/\delta)}{\Delta_a} - \sigma^2 \sigma_0^{-2} \Delta_a\right) \mathbb{1}\{\Delta_a \geq \varepsilon\}\right] . \tag{7}$$

**Case 2: The gap is small, $\Delta_{A_t} < \varepsilon$.** Then naively $\sum_{t=1}^n \mathbb{E}\left[\Delta_{A_t} \mathbb{1}\{\Delta_{A_t} < \varepsilon\}\right] < \varepsilon n$.

**Case 3: Event $E_t$ does not occur.** The last term in (5) can be bounded as

$$\mathbb{E}\left[\Delta_{A_t} \mathbb{1}\{\bar{E}_t\}\right] \tag{8}$$
$$= \mathbb{E}\left[\mathbb{E}\left[(\theta_{A_*} - \theta_{A_t}) \mathbb{1}\{\bar{E}_t\} \mid H_t\right]\right]$$
$$\leq \mathbb{E}\left[\mathbb{E}\left[(\theta_{A_*} - U_{t,A_*}) \mathbb{1}\{\bar{E}_t\} \mid H_t\right] + \mathbb{E}\left[(U_{t,A_t} - \theta_{A_t}) \mathbb{1}\{\bar{E}_t\} \mid H_t\right]\right]$$
$$\leq \mathbb{E}\left[\mathbb{E}\left[(\theta_{A_*} - \hat{\theta}_{t,A_*}) \mathbb{1}\{\bar{E}_t\} \mid H_t\right] + \mathbb{E}\left[(\hat{\theta}_{t,A_t} - \theta_{A_t}) \mathbb{1}\{\bar{E}_t\} \mid H_t\right] + \mathbb{E}\left[C_{t,A_t} \mathbb{1}\{\bar{E}_t\} \mid H_t\right]\right] .$$

To bound the resulting terms, we use that $\theta_a - \hat{\theta}_{t,a} \mid H_t \sim \mathcal{N}(0, \hat{\sigma}_{t,a}^2)$.

**Lemma 7.** *For any action $a \in \mathcal{A}$, round $t \in [n]$, and history $H_t$,*

$$\mathbb{E}\left[(\theta_a - \hat{\theta}_{t,a}) \mathbb{1}\{\bar{E}_t\} \mid H_t\right] \leq 2\sigma_0 K\delta .$$

*Proof.* Let $E_{t,a} = \left\{ |\theta_a - \hat{\theta}_{t,a}| \leq C_{t,a} \right\}$. We start with decomposing $\bar{E}_t$ into individual $\bar{E}_{t,a}$ as

$$\mathbb{E}\left[ (\theta_a - \hat{\theta}_{t,a}) \mathbb{1}\{\bar{E}_t\} \,\Big|\, H_t \right] \leq \mathbb{E}\left[ |\theta_a - \hat{\theta}_{t,a}| \mathbb{1}\{\bar{E}_{t,a}\} \,\Big|\, H_t \right] + \sum_{a' \neq a} \mathbb{E}\left[ |\theta_a - \hat{\theta}_{t,a}| \mathbb{1}\{\bar{E}_{t,a'}\} \,\Big|\, H_t \right] .$$

To bound the first term, we use that $\theta_a - \hat{\theta}_{t,a} \mid H_t \sim \mathcal{N}(0, \hat{\sigma}_{t,a}^2)$. Thus

$$\mathbb{E}\left[ |\theta_a - \hat{\theta}_{t,a}| \mathbb{1}\{\bar{E}_{t,a}\} \,\Big|\, H_t \right] \leq \frac{2}{\sqrt{2\pi \hat{\sigma}_{t,a}^2}} \int_{x = C_{t,a}}^{\infty} x \exp\left[ -\frac{x^2}{2\hat{\sigma}_{t,a}^2} \right] \mathrm{d}x$$

$$= -\sqrt{\frac{2\hat{\sigma}_{t,a}^2}{\pi}} \int_{x = C_{t,a}}^{\infty} \frac{\partial}{\partial x} \left( \exp\left[ -\frac{x^2}{2\hat{\sigma}_{t,a}^2} \right] \right) \mathrm{d}x$$

$$= \sqrt{\frac{2\hat{\sigma}_{t,a}^2}{\pi}} \delta \leq \sigma_0 \delta .$$

To bound the second term, we use the independence of the distributions for $a$ and $a'$,

$$\mathbb{E}\left[ |\theta_a - \hat{\theta}_{t,a}| \mathbb{1}\{\bar{E}_{t,a'}\} \,\Big|\, H_t \right] = \mathbb{E}\left[ |\theta_a - \hat{\theta}_{t,a}| \,\Big|\, H_t \right] \mathbb{P}\left( \bar{E}_{t,a'} \mid H_t \right) .$$

The probability is at most $2\delta$ and the expectation can be bounded as

$$\mathbb{E}\left[ |\theta_a - \hat{\theta}_{t,a}| \,\Big|\, H_t \right] = \mathbb{E}\left[ \sqrt{(\theta_a - \hat{\theta}_{t,a})^2} \,\Big|\, H_t \right] \leq \sqrt{\mathbb{E}\left[ (\theta_a - \hat{\theta}_{t,a})^2 \,\Big|\, H_t \right]} = \hat{\sigma}_{t,a} \leq \sigma_0 .$$

This completes the proof. $\qquad\square$

The first two terms in (8) can be bounded using a union bound over $a \in \mathcal{A}$ and Lemma 7. For the last term, we use that $C_{t,a} \leq \sqrt{2\sigma_0^2 \log(1/\delta)}$ and a union bound in $\mathbb{P}\left( \bar{E}_t \mid H_t \right)$ to get

$$\mathbb{E}\left[ C_{t,A_t} \mathbb{1}\{\bar{E}_t\} \mid H_t \right] \leq \sqrt{2\sigma_0^2 \log(1/\delta)} \mathbb{P}\left( \bar{E}_t \mid H_t \right) \leq 2\sqrt{2 \log(1/\delta)} \sigma_0 K \delta .$$

Finally, we sum up the upper bounds on (8) over all rounds $t \in [n]$.

## A.2 Proof of Corollary 2

The key idea in the proof is to integrate out the random gap in (7). Fix action $a \in \mathcal{A}$ and thresholds $\varepsilon_2 > \varepsilon > 0$. We consider two cases.

**Case 1: Diminishing gaps $\varepsilon < \Delta_a \leq \varepsilon_2$.** Let

$$\xi_a(\theta_a^*) = \arg\max_{x \in [\varepsilon, \varepsilon_2]} h_a(\theta_a^* - x)$$

and $N_a$ be defined as in Appendix A.1. Then

$$\mathbb{E}\left[ \Delta_a N_a \mathbb{1}\{\varepsilon < \Delta_a < \varepsilon_2\} \right] = \int_{\theta_{-a}} \int_{\theta_a = \theta_a^* - \varepsilon_2}^{\theta_a^* - \varepsilon} \Delta_a N_a h_a(\theta_a) \, \mathrm{d}\theta_a \, h_{-a}(\theta_{-a}) \, \mathrm{d}\theta_{-a}$$

$$\leq \int_{\theta_{-a}} \left( \int_{\theta_a = \theta_a^* - \varepsilon_2}^{\theta_a^* - \varepsilon} \Delta_a N_a \, \mathrm{d}\theta_a \right) h_a(\theta_a^* - \xi_a(\theta_a^*)) \, h_{-a}(\theta_{-a}) \, \mathrm{d}\theta_{-a} ,$$

where the inequality is by the definition of $\xi_a$. Now the inner integral is independent of $h_a$ and thus can be easily bounded. Specifically, the upper bound in (6) and simple integration yield

$$\int_{\theta_a = \theta_a^* - \varepsilon_2}^{\theta_a^* - \varepsilon} \Delta_a N_a \, \mathrm{d}\theta_a \leq \int_{\theta_a = \theta_a^* - \varepsilon_2}^{\theta_a^* - \varepsilon} \left( \frac{8\sigma^2 \log(1/\delta)}{\theta_a^* - \theta_a} - \sigma^2 \sigma_0^{-2} (\theta_a^* - \theta_a) \right) \mathrm{d}\theta_a$$

$$= 8\sigma^2 \log(1/\delta)(\log \varepsilon_2 - \log \varepsilon) - \frac{\sigma^2 (\varepsilon_2^2 - \varepsilon^2)}{2\sigma_0^2} .$$

For $\varepsilon = 1/n$ and $\varepsilon_2 = 1/\sqrt{\log n}$, we get

$$\int_{\theta_a = \theta_a^* - \varepsilon_2}^{\theta_a^* - \varepsilon} \Delta_a N_a \, \mathrm{d}\theta_a \leq 8\sigma^2 \log(1/\delta) \log n - \frac{\sigma^2}{2\sigma_0^2 \log n} + \frac{\sigma^2}{2\sigma_0^2 n^2} - 4\sigma^2 \log(1/\delta) \log \log n \quad (9)$$

$$\leq 8\sigma^2 \log(1/\delta) \log n - \frac{\sigma^2}{2\sigma_0^2 \log n} \; .$$

The last inequality holds for $\sigma_0^2 \geq \frac{1}{8 \log(1/\delta) \, n^2 \log \log n}$.

**Case 2: Large gaps $\Delta_a > \varepsilon_2$.** Here we use (6) together with $\varepsilon_2 = 1/\sqrt{\log n}$ to get

$$\mathbb{E}\left[\Delta_a N_a \mathbb{1}\{\Delta_a > \varepsilon_2\}\right] \leq \mathbb{E}\left[\frac{8\sigma^2 \log(1/\delta)}{\Delta_a} \mathbb{1}\{\Delta_a > \varepsilon_2\}\right] < 8\sigma^2 \log(1/\delta) \sqrt{\log n} \; . \quad (10)$$

Finally, we chain all inequalities.

### A.3 Proof of Lemma 3

We have that

$$\sum_{a \in \mathcal{A}} \int_{\theta_{-a}} h_a(\theta_a^*) \, h_{-a}(\theta_{-a}) \, \mathrm{d}\theta_{-a}$$

$$\leq \sum_{a \in \mathcal{A}} \int_{\theta_{-a}} \left(\sum_{a' \neq a} h_a(\theta_{a'})\right) \left(\prod_{a' \neq a} h_{a'}(\theta_{a'})\right) \mathrm{d}\theta_{-a}$$

$$= \sum_{a \in \mathcal{A}} \sum_{a' \neq a} \int_{\theta_{a'}} h_a(\theta_{a'}) \, h_{a'}(\theta_{a'}) \, \mathrm{d}\theta_{a'}$$

$$= \frac{1}{2\pi\sigma_0^2} \sum_{a \in \mathcal{A}} \sum_{a' \neq a} \int_{\theta_{a'}} \exp\left[-\frac{(\theta_{a'} - \mu_{0,a})^2}{2\sigma_0^2} - \frac{(\theta_{a'} - \mu_{0,a'})^2}{2\sigma_0^2}\right] \mathrm{d}\theta_{a'}$$

$$= \frac{1}{2\sqrt{\pi\sigma_0^2}} \sum_{a \in \mathcal{A}} \sum_{a' \neq a} \exp\left[-\frac{(\mu_{0,a} - \mu_{0,a'})^2}{4\sigma_0^2}\right] \; ,$$

where the last step is by completing the square and integrating out $\theta_{a'}$.

### A.4 Proof of Theorem 4

The regret bound of UCB1 is proved similarly to Theorem 1 and Corollary 2. This is because UCB1 can be viewed as BayesUCB where $\sigma_0 = \infty$ and each action $a \in \mathcal{A}$ is initially taken once at round $t = a$. Since $\sigma_0 = \infty$, the confidence interval becomes

$$C_{t,a} = \sqrt{\frac{2\sigma^2 \log(1/\delta)}{N_{t,a}}} \; .$$

The proof differs in two steps. First, the regret in the first $K$ rounds is bounded by $\sum_{a \in \mathcal{A}} \mathbb{E}[\Delta_a]$. Second, the concentration argument (Case 3 in Appendix A.1) changes because the bandit instance $\theta$ is fixed and the estimated model parameter $\hat{\theta}_t$ is random. We detail it below.

**Case 3: Event $E_t$ does not occur.** The last term in (5) can be bounded as

$$\mathbb{E}\left[\Delta_{A_t} \mathbb{1}\{\bar{E}_t\}\right] \quad (11)$$

$$= \mathbb{E}\left[\mathbb{E}\left[(\theta_{A_*} - \theta_{A_t}) \mathbb{1}\{\bar{E}_t\} \,\middle|\, \theta\right]\right]$$

$$\leq \mathbb{E}\left[\mathbb{E}\left[(\theta_{A_*} - U_{t,A_*}) \mathbb{1}\{\bar{E}_t\} \,\middle|\, \theta\right] + \mathbb{E}\left[(U_{t,A_t} - \theta_{A_t}) \mathbb{1}\{\bar{E}_t\} \,\middle|\, \theta\right]\right]$$

$$\leq \mathbb{E}\left[\mathbb{E}\left[(\theta_{A_*} - \hat{\theta}_{t,A_*}) \mathbb{1}\{\bar{E}_t\} \,\middle|\, \theta\right] + \mathbb{E}\left[(\hat{\theta}_{t,A_t} - \theta_{A_t}) \mathbb{1}\{\bar{E}_t\} \,\middle|\, \theta\right] + \mathbb{E}\left[C_{t,A_t} \mathbb{1}\{\bar{E}_t\} \,\middle|\, \theta\right]\right] \; .$$

To bound the resulting terms, we use that $\theta_a - \hat{\theta}_{t,a} \mid \theta \sim \mathcal{N}(0, \sigma^2/N_{t,a})$.

**Lemma 8.** *For any action $a \in \mathcal{A}$, round $t > K$, and $N_{t,a} \geq 1$,*

$$\mathbb{E}\left[(\theta_a - \hat{\theta}_{t,a})\mathbb{1}\{\bar{E}_t\} \,\middle|\, \theta\right] \leq 2\sigma K\delta \,.$$

*Proof.* Let $E_{t,a} = \left\{|\theta_a - \hat{\theta}_{t,a}| \leq C_{t,a}\right\}$. We start with decomposing $\bar{E}_t$ into individual $\bar{E}_{t,a}$ as

$$\mathbb{E}\left[(\theta_a - \hat{\theta}_{t,a})\mathbb{1}\{\bar{E}_t\} \,\middle|\, \theta\right] \leq \mathbb{E}\left[|\theta_a - \hat{\theta}_{t,a}|\mathbb{1}\{\bar{E}_{t,a}\} \,\middle|\, \theta\right] + \sum_{a' \neq a} \mathbb{E}\left[|\theta_a - \hat{\theta}_{t,a}|\mathbb{1}\{\bar{E}_{t,a'}\} \,\middle|\, \theta\right] \,.$$

To bound the first term, we use that $\theta_a - \hat{\theta}_{t,a} \mid \theta \sim \mathcal{N}(0, \sigma^2/N_{t,a})$. Thus

$$\mathbb{E}\left[|\theta_a - \hat{\theta}_{t,a}|\mathbb{1}\{\bar{E}_t\} \,\middle|\, \theta\right] \leq \frac{2}{\sqrt{2\pi\sigma^2/N_{t,a}}} \int_{x=C_{t,a}}^{\infty} x \exp\left[-\frac{x^2}{2\sigma^2/N_{t,a}}\right] \mathrm{d}x$$

$$= -\sqrt{\frac{2\sigma^2}{\pi N_{t,a}}} \int_{x=C_{t,a}}^{\infty} \frac{\partial}{\partial x}\left(\exp\left[-\frac{x^2}{2\sigma^2/N_{t,a}}\right]\right) \mathrm{d}x$$

$$= \sqrt{\frac{2\sigma^2}{\pi N_{t,a}}}\delta \leq \sigma\delta \,.$$

To bound the second term, we use the independence of the distributions for $a$ and $a'$,

$$\mathbb{E}\left[|\theta_a - \hat{\theta}_{t,a}|\mathbb{1}\{\bar{E}_{t,a'}\} \,\middle|\, \theta\right] = \mathbb{E}\left[|\theta_a - \hat{\theta}_{t,a}| \,\middle|\, \theta\right] \mathbb{P}\left(\bar{E}_{t,a'} \,\middle|\, \theta\right) \,.$$

The probability is at most $2\delta$ and the expectation can be bounded as

$$\mathbb{E}\left[|\theta_a - \hat{\theta}_{t,a}| \,\middle|\, \theta\right] = \mathbb{E}\left[\sqrt{(\theta_a - \hat{\theta}_{t,a})^2} \,\middle|\, \theta\right] \leq \sqrt{\mathbb{E}\left[(\theta_a - \hat{\theta}_{t,a})^2 \,\middle|\, \theta\right]} = \sqrt{\frac{\sigma^2}{N_{t,a}}} \leq \sigma \,.$$

This completes the proof. $\qquad\square$

The first two terms in (11) can be bounded using a union bound over $a \in \mathcal{A}$ and Lemma 8. For the last term, we use that $C_{t,a} \leq \sqrt{2\sigma^2 \log(1/\delta)}$ and a union bound in $\mathbb{P}\left(\bar{E}_t \mid \theta\right)$ to get

$$\mathbb{E}\left[C_{t,A_t}\mathbb{1}\{\bar{E}_t\} \,\middle|\, \theta\right] \leq \sqrt{2\sigma^2 \log(1/\delta)}\mathbb{P}\left(\bar{E}_t \,\middle|\, \theta\right) \leq 2\sqrt{2\log(1/\delta)}\sigma K\delta \,.$$

Finally, we sum up the upper bounds on (11) over all rounds $t \in [n]$.

### A.5 Proof of Theorem 5

Let $E_t = \left\{\forall a \in \mathcal{A} : |\theta_a - \hat{\theta}_{t,a}| \leq C_{t,a}\right\}$ be the event that all confidence intervals at round $t$ hold. Fix $\varepsilon > 0$. We decompose the $n$-round regret as in (5) and then bound each resulting term next.

**Case 1: Event $E_t$ occurs and the gap is large, $\Delta_{A_t} \geq \varepsilon$.** As in Appendix A.1,

$$\Delta_{A_t} = \theta_{A_*} - \theta_{A_t} \leq \theta_{A_*} - U_{t,A_*} + U_{t,A_t} - \theta_{A_t} \leq U_{t,A_t} - \theta_{A_t} \leq 2C_{t,A_t} \,.$$

In the second inequality, we use that $\theta_{A_*} \leq U_{t,A_*}$ on event $E_t$. This implies that on event $E_t$, action $a$ can be taken only if

$$N_{t,a} \leq \frac{2\log(1/\delta)}{\Delta_a^2} - (\alpha_a + \beta_a + 1) \,.$$

Now we apply this inequality to bound the first term in (5) as

$$\sum_{t=1}^{n} \mathbb{E}\left[\Delta_{A_t}\mathbb{1}\{\Delta_{A_t} \geq \varepsilon, E_t\}\right] \leq \mathbb{E}\left[\sum_{a \neq A_*} \left(\frac{2\log(1/\delta)}{\Delta_a} - (\alpha_a + \beta_a + 1)\Delta_a\right)\mathbb{1}\{\Delta_a \geq \varepsilon\}\right] \,.$$

**Case 2: The gap is small, $\Delta_{A_t} < \varepsilon$.** Then naively $\sum_{t=1}^{n} \mathbb{E}\left[\Delta_{A_t}\mathbb{1}\{\Delta_{A_t} < \varepsilon\}\right] < \varepsilon n$.

**Case 3: Event $E_t$ does not occur.** Since $\theta_a \in [0, 1]$, the last term in (5) can be bounded as

$$\mathbb{E}\left[\Delta_{A_t}\mathbb{1}\{\bar{E}_t\}\right] \leq \mathbb{E}\left[\mathbb{P}\left(\bar{E}_t \mid H_t\right)\right] \leq 2K\delta.$$

This completes the first part of the proof.

The second claim is proved as in Appendix A.2 and we only comment on what differs. For $\varepsilon = 1/n$ and $\varepsilon_2 = 1/\sqrt{\log n}$, (9) becomes

$$\begin{aligned}
\int_{\theta_a = \theta_a^* - \varepsilon_2}^{\theta_a^* - \varepsilon} \Delta_a N_a \, \mathrm{d}\theta_a &\leq \int_{\theta_a = \theta_a^* - \varepsilon_2}^{\theta_a^* - \varepsilon} \frac{2\log(1/\delta)}{\theta_a^* - \theta_a} - \lambda(\theta_a^* - \theta_a) \, \mathrm{d}\theta_a \\
&= 2\log(1/\delta)(\log \varepsilon_2 - \log \varepsilon) - \frac{\lambda(\varepsilon_2^2 - \varepsilon^2)}{2} \\
&= 2\log(1/\delta)\log n - \frac{\lambda}{2\log n} + \frac{\lambda}{2n^2} - \log(1/\delta)\log\log n \\
&\leq 2\log(1/\delta)\log n - \frac{\lambda}{2\log n}.
\end{aligned}$$

The last inequality holds for $\lambda \leq 2\log(1/\delta)\, n^2 \log\log n$. Moreover, (10) becomes

$$\mathbb{E}\left[\Delta_a N_a \mathbb{1}\{\Delta_a > \varepsilon_2\}\right] \leq \mathbb{E}\left[\frac{2\log(1/\delta)}{\Delta_a}\mathbb{1}\{\Delta_a > \varepsilon_2\}\right] < 2\log(1/\delta)\sqrt{\log n}.$$

This completes the second part of the proof.

## A.6 Proof of Theorem 6

Let

$$E_t = \left\{\forall a \in \mathcal{A} : |a^\top(\theta - \hat{\theta}_t)| \leq \sqrt{2\log(1/\delta)}\|a\|_{\hat{\Sigma}_t}\right\} \tag{12}$$

be an event that high-probability confidence intervals for mean rewards at round $t$ hold. Our proof has three parts.

**Case 1: Event $E_t$ occurs and the gap is large, $\Delta_{A_t} \geq \varepsilon$.** Then

$$\begin{aligned}
\Delta_{A_t} &= \frac{1}{\Delta_{A_t}}\Delta_{A_t}^2 \leq \frac{1}{\Delta_{\min}^\varepsilon}(A_*^\top\theta - A_t^\top\theta)^2 \leq \frac{1}{\Delta_{\min}^\varepsilon}(A_*^\top\theta - U_{t,A_*} + U_{t,A_t} - A_t^\top\theta)^2 \\
&\leq \frac{1}{\Delta_{\min}^\varepsilon}(U_{t,A_t} - A_t^\top\theta)^2 \leq \frac{4}{\Delta_{\min}^\varepsilon}C_{t,A_t}^2 = \frac{8\log(1/\delta)}{\Delta_{\min}^\varepsilon}\|A_t\|_{\hat{\Sigma}_t}^2.
\end{aligned}$$

The first inequality follows from definitions of $\Delta_{A_t}$ and $\Delta_{\min}^\varepsilon$; and that the gap is large, $\Delta_{A_t} \geq \varepsilon$. The second inequality holds because $A_*^\top\theta - A_t^\top\theta \geq 0$ by definition and $U_{t,A_t} - U_{t,A_*} \geq 0$ by the design of BayesUCB. The third inequality holds because $A_*^\top\theta - U_{t,A_*} \leq 0$ on event $E_t$. Specifically, for any action $a \in \mathcal{A}$ on event $E_t$,

$$a^\top\theta - U_{t,a} = a^\top(\theta - \hat{\theta}_t) - C_{t,a} \leq C_{t,a} - C_{t,a} = 0.$$

The last inequality follows from the definition of event $E_t$. Specifically, for any action $a \in \mathcal{A}$ on event $E_t$,

$$U_{t,a} - a^\top\theta = a^\top(\hat{\theta}_t - \theta) + C_{t,a} \leq C_{t,a} + C_{t,a} = 2C_{t,a}.$$

**Case 2: The gap is small, $\Delta_{A_t} \leq \varepsilon$.** Then naively $\Delta_{A_t} \leq \varepsilon$.

**Case 3: Event $E_t$ does not occur.** Then $\Delta_{A_t}\mathbb{1}\{\bar{E}_t\} \leq 2\|A_t\|_2\|\theta\|_2\mathbb{1}\{\bar{E}_t\} \leq 2LL_*\mathbb{1}\{\bar{E}_t\}$, where $2LL_*$ is a trivial upper bound on $\Delta_{A_t}$. We bound the event in expectation as follows.

**Lemma 9.** *For any round $t \in [n]$ and history $H_t$, we have that $\mathbb{P}\left(\bar{E}_t \mid H_t\right) \leq 2K\delta$.*

*Proof.* First, note that for any history $H_t$,

$$\mathbb{P}\left(\bar{E}_t \mid H_t\right) \leq \sum_{a \in \mathcal{A}} \mathbb{P}\left(|a^\top(\theta - \hat{\theta}_t)| \geq \sqrt{2\log(1/\delta)}\|a\|_{\hat{\Sigma}_t} \;\middle|\; H_t\right).$$

By definition, $\theta - \hat{\theta}_t \mid H_t \sim \mathcal{N}(\mathbf{0}_d, \hat{\Sigma}_t)$, and therefore $a^\top(\theta - \hat{\theta}_t)/\|a\|_{\hat{\Sigma}_t} \mid H_t \sim \mathcal{N}(0,1)$ for any action $a \in \mathcal{A}$. It immediately follows that

$$\mathbb{P}\left(|a^\top(\theta - \hat{\theta}_t)| \geq \sqrt{2\log(1/\delta)}\|a\|_{\hat{\Sigma}_t} \;\Big|\; H_t\right) \leq 2\delta\,.$$

This completes the proof. $\qquad\square$

Finally, we chain all inequalities, add them over all rounds, and get

$$R(n) \leq 8\mathbb{E}\left[\frac{1}{\Delta_{\min}^\varepsilon}\sum_{t=1}^n \|A_t\|_{\hat{\Sigma}_t}^2\right]\log(1/\delta) + \varepsilon n + 4LL_* Kn\delta\,.$$

The sum can bounded using a worst-case argument below, which yields our claim.

**Lemma 10.** *The sum of posterior variances is bounded as*

$$\sum_{t=1}^n \|A_t\|_{\hat{\Sigma}_t}^2 \leq \frac{\sigma_{0,\max}^2 d}{\log\left(1 + \frac{\sigma_{0,\max}^2}{\sigma^2}\right)}\log\left(1 + \frac{\sigma_{0,\max}^2 n}{\sigma^2 d}\right)\,.$$

*Proof.* We start with an upper bound on the posterior variance of the mean reward estimate of any action. In any round $t \in [n]$, by Weyl's inequalities, we have

$$\lambda_1(\hat{\Sigma}_t) = \lambda_1((\Sigma_0^{-1} + G_t)^{-1}) = \lambda_d^{-1}(\Sigma_0^{-1} + G_t) \leq \lambda_d^{-1}(\Sigma_0^{-1}) = \lambda_1(\Sigma_0)\,.$$

Thus, when $\|a\|_2 \leq L$ for any action $a \in \mathcal{A}$, we have $\max_{a \in \mathcal{A}}\|a\|_{\hat{\Sigma}_t} \leq \sqrt{\lambda_1(\Sigma_0)}L = \sigma_{0,\max}$.

Now we bound the sum of posterior variances $\sum_{t=1}^n \|A_t\|_{\hat{\Sigma}_t}^2$. Fix round $t$ and note that

$$\|A_t\|_{\hat{\Sigma}_t}^2 = \sigma^2 \frac{A_t^\top \hat{\Sigma}_t A_t}{\sigma^2} \leq c_1 \log(1 + \sigma^{-2} A_t^\top \hat{\Sigma}_t A_t) = c_1 \log\det(I_d + \sigma^{-2}\hat{\Sigma}_t^{\frac{1}{2}} A_t A_t^\top \hat{\Sigma}_t^{\frac{1}{2}}) \quad (13)$$

for

$$c_1 = \frac{\sigma_{0,\max}^2}{\log(1 + \sigma^{-2}\sigma_{0,\max}^2)}\,.$$

This upper bound is derived as follows. For any $x \in [0, u]$,

$$x = \frac{x}{\log(1+x)}\log(1+x) \leq \left(\max_{x \in [0,u]}\frac{x}{\log(1+x)}\right)\log(1+x) = \frac{u}{\log(1+u)}\log(1+x)\,.$$

Then we set $x = \sigma^{-2}A_t^\top \hat{\Sigma}_t A_t$ and use the definition of $\sigma_{0,\max}$.

The next step is bounding the logarithmic term in (13), which can be rewritten as

$$\log\det(I_d + \sigma^{-2}\hat{\Sigma}_t^{\frac{1}{2}}A_t A_t^\top \hat{\Sigma}_t^{\frac{1}{2}}) = \log\det(\hat{\Sigma}_t^{-1} + \sigma^{-2}A_t A_t^\top) - \log\det(\hat{\Sigma}_t^{-1})\,.$$

Because of that, when we sum over all rounds, we get telescoping and the total contribution of all terms is at most

$$\sum_{t=1}^n \log\det(I_d + \sigma^{-2}\hat{\Sigma}_t^{\frac{1}{2}}A_t A_t^\top \hat{\Sigma}_t^{\frac{1}{2}}) = \log\det(\hat{\Sigma}_{n+1}^{-1}) - \log\det(\hat{\Sigma}_1^{-1})$$

$$= \log\det(\Sigma_0^{\frac{1}{2}}\hat{\Sigma}_{n+1}^{-1}\Sigma_0^{\frac{1}{2}})$$

$$\leq d\log\left(\frac{1}{d}\mathrm{tr}(\Sigma_0^{\frac{1}{2}}\hat{\Sigma}_{n+1}^{-1}\Sigma_0^{\frac{1}{2}})\right)$$

$$= d\log\left(1 + \frac{1}{\sigma^2 d}\sum_{t=1}^n \mathrm{tr}(\Sigma_0^{\frac{1}{2}}A_t A_t^\top \Sigma_0^{\frac{1}{2}})\right)$$

$$= d\log\left(1 + \frac{1}{\sigma^2 d}\sum_{t=1}^n A_t^\top \Sigma_0 A_t\right)$$

$$\leq d\log\left(1 + \frac{\sigma_{0,\max}^2 n}{\sigma^2 d}\right)\,.$$

This completes the proof. $\qquad\square$

# B   Complete Statement of Corollary 2

**Theorem 11.** *Let $\sigma_0^2 \geq \frac{1}{8\log(1/\delta)\, n^2 \log\log n}$. Then there exist functions $\xi_a : \mathbb{R} \to \left[\frac{1}{n}, \frac{1}{\sqrt{\log n}}\right]$ such that the $n$-round Bayes regret of* `BayesUCB` *in a $K$-armed Gaussian bandit is bounded as*

$$R(n) \leq \left[8\sigma^2 \log(1/\delta)\log n - \frac{\sigma^2}{2\sigma_0^2 \log n}\right] \sum_{a\in\mathcal{A}} \int_{\theta_{-a}} h_a(\theta_a^* - \xi_a(\theta_a^*))\, h_{-a}(\theta_{-a})\, \mathrm{d}\theta_{-a} + C\,,$$

*where $C = 8\sigma^2 K \log(1/\delta)\sqrt{\log n} + (2\sqrt{2\log(1/\delta)}+1)\sigma_0 K n \delta + 1$ is a low-order term.*

*Moreover, when $\sigma_0^2 < \frac{1}{8\log(1/\delta)\, n^2 \log\log n}$, the regret is bounded as*

$$R(n) \leq \frac{2\sqrt{2\log(1/\delta)}+1}{\sqrt{8\log(1/\delta)\log\log n}} K\delta + 1\,.$$

*Proof.* The first claim is proved in Appendix A.2. The second claim can be proved as follows. Take Theorem 1, set $\varepsilon = 0$, and consider the three cases in Appendix A.1.

**Case 1: Event $E_t$ occurs and the gap is large, $\Delta_{A_t} \geq \varepsilon$.** On event $E_t$, action $a$ can be taken only if

$$\Delta_a \leq 2\sqrt{\frac{2\log(1/\delta)}{\sigma_0^{-2} + \sigma^{-2}N_{t,a}}} \leq 2\sqrt{2\sigma_0^2 \log(1/\delta)} \leq 2\sqrt{\frac{1}{4n^2 \log\log n}} < \frac{1}{n}\,.$$

Therefore, the corresponding $n$-round regret is bounded by 1.

**Case 2: The gap is small, $\Delta_{A_t} < \varepsilon$.** This case cannot happen because $\varepsilon = 0$.

**Case 3: Event $E_t$ does not occur.** The $n$-round regret is bounded by

$$(2\sqrt{2\log(1/\delta)}+1)\sigma_0 K n \delta \leq \frac{2\sqrt{2\log(1/\delta)}+1}{\sqrt{8\log(1/\delta)\log\log n}} K\delta\,.$$

This completes the proof. □

# C   Gap-Free Regret Bound of `BayesUCB` in Linear Bandit

Let $E_t$ be the event in (12). Our proof has three parts.

**Case 1: Event $E_t$ occurs and the gap is large, $\Delta_{A_t} \geq \varepsilon$.** Then

$$\Delta_{A_t} = A_*^\top \theta - A_t^\top \theta \leq A_*^\top \theta - U_{t,A_*} + U_{t,A_t} - A_t^\top \theta \leq U_{t,A_t} - A_t^\top \theta \leq 2C_{t,A_t}\,.$$

The first inequality holds because $U_{t,A_t} - U_{t,A_*} \geq 0$ by the design of `BayesUCB`. The second one uses that $A_*^\top \theta - U_{t,A_*} \leq 0$. Specifically, for any action $a \in \mathcal{A}$ on event $E_t$,

$$a^\top \theta - U_{t,a} = a^\top(\theta - \hat{\theta}_t) - C_{t,a} \leq C_{t,a} - C_{t,a} = 0\,.$$

The last inequality follows from the definition of event $E_t$. Specifically, for any action $a \in \mathcal{A}$ on event $E_t$,

$$U_{t,a} - a^\top \theta = a^\top(\hat{\theta}_t - \theta) + C_{t,a} \leq C_{t,a} + C_{t,a} = 2C_{t,a}\,.$$

**Cases 2 and 3** are bounded as in Appendix A.6. Now we chain all inequalities, add them over all rounds, and get

$$R(n) \leq 2\mathbb{E}\left[\sum_{t=1}^n \|A_t\|_{\hat{\Sigma}_t}\right]\sqrt{2\log(1/\delta)} + \varepsilon n + 4LL_* K n \delta$$

$$\leq 2\sqrt{\mathbb{E}\left[\sum_{t=1}^n \|A_t\|_{\hat{\Sigma}_t}^2\right]}\sqrt{2n\log(1/\delta)} + \varepsilon n + 4LL_* K n \delta\,,$$

where the last inequality uses the Cauchy-Schwarz inequality and the concavity of the square root. Finally, the sum $\sum_{t=1}^n \|A_t\|_{\hat{\Sigma}_t}^2$ is bounded using Lemma 10. This completes the proof.

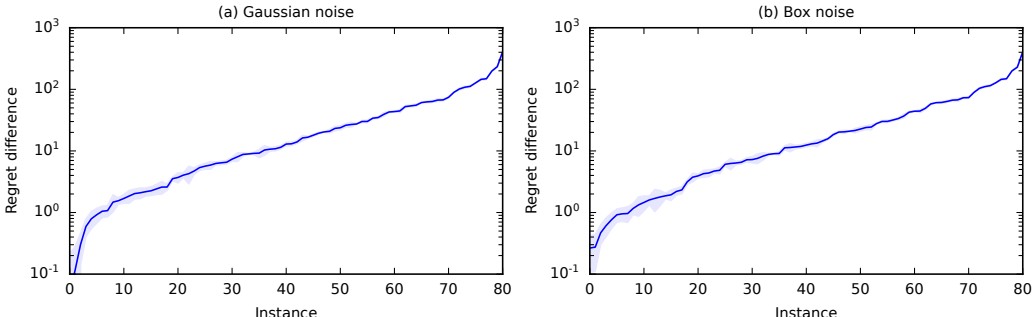

Figure 3: The difference in regret of `UCB1` and `BayesUCB` on 81 Bayesian bandit instances, sorted by the difference. In plot (a), the noise is Gaussian $\mathcal{N}(0, \sigma^2)$. In plot (b), the noise is $\sigma$ with probability $0.5$ and $-\sigma$ otherwise.

## D    Comparison of `BayesUCB` and `UCB1`

We report the difference in regret of `UCB1` and `BayesUCB` on 81 Bayesian bandit instances. These instances are obtained by all combinations of $K \in \{5, 10, 20\}$ actions, reward noise $\sigma \in \{0.5, 1, 2\}$, prior gap $\Delta_0 \in \{0.5, 1, 2\}$, and prior width $\sigma_0 \in \{0.5, 1, 2\}$. The horizon is $n = 1\,000$ rounds and all results are averaged over $1\,000$ random runs.

Our results are reported in Figure 3. In Figure 3a, the noise is Gaussian $\mathcal{N}(0, \sigma^2)$. In Figure 3b, the noise is $\sigma$ with probability $0.5$ and $-\sigma$ otherwise. Therefore, this noise is $\sigma^2$-sub-Gaussian, of the same magnitude as $\mathcal{N}(0, \sigma^2)$ but far from it in terms of the distribution. This tests the robustness of `BayesUCB` to Gaussian posterior updates. `UCB1` only needs $\sigma^2$-sub-Gaussian noise. In both plots, and in all 81 Bayesian bandit instances, `BayesUCB` has a lower regret than `UCB1`. It is also remarkably robust to noise misspecification, although we cannot prove it.

## E    Note on Information-Theory Bounds

Our approach could be used to derive Bayesian information-theory bounds [Russo and Van Roy, 2016]. The key step in these bounds, where the information-theory term $I_{t,a}$ for action $a$ at round $t$ arises, is $\Delta_{A_t} \leq \Gamma \sqrt{I_{t,A_t}}$, where $\Gamma$ is the highest possible ratio of regret to information gain. As in Case 1 in Appendix A.6, the $n$-round regret can be bounded as

$$\sum_{t=1}^{n} \Delta_{A_t} = \sum_{t=1}^{n} \frac{1}{\Delta_{A_t}} \Delta_{A_t}^2 \leq \frac{1}{\Delta_{\min}^{\varepsilon}} \sum_{t=1}^{n} \Delta_{A_t}^2 \leq \frac{\Gamma^2}{\Delta_{\min}^{\varepsilon}} \sum_{t=1}^{n} I_{t,A_t} \,.$$

The term $\sum_{t=1}^{n} I_{t,A_t}$ can be bounded using a worst-case argument by a $O(\log n)$ bound.

