# OpenReview forum: "Finite-Time Logarithmic Bayes Regret Upper Bounds"
_NeurIPS.cc/2023/Conference — NeurIPS 2023 poster_

### Official Review · Reviewer_Q5oQ · 2023-06-25

**Soundness:** 3 good
**Presentation:** 3 good
**Contribution:** 3 good
**Rating:** 5
**Confidence:** 3

**Summary:**

This paper proves logarithmic (in horizon $n$) regret bounds for Bayesian bandits using the BayesUCB algorithm. They specifically study the settings of Gaussian MABs and Gaussian linear bandits. This improves on prior works which show the more standard $\sqrt{n}$ bounds on regret.

**Strengths:**

- Generally the writing is good. Problem setup and results are clear, as well as points of comparison with classical asymptotic guarantees.
- The proofs are simple and easy to understand.
- Numerical simulations are run to back up the validity of the theory.

**Weaknesses:**

- On the technical side, I'm not sure where the contributions are. The abstract states that "our proofs mark a technical departure from prior works". Can this be made more explicit in the technical discussion?
- Several questions about the experiments, see next section. Generally the scope of the experiments is rather limited (which may be fine since this is primarily a theoretical work).

Some comments on writing:
- line 65: "any bounded $\theta$": since we have not specified a specific parametric model here (it is not even clear that $\theta$ must lie in some Euclidean space) a description of $\theta$ being bounded seems premature here.
- as a matter of writing style, it is a bit strange to put part of the inequality inline and the other half in display equations (such as line 120).

**Questions:**


1. Can the authors elaborate on how the techniques in the paper differ from prior works, and/or are clearer and simpler to understand?

    a) From my understanding, the main parts of the analysis seem pretty similar to the standard frequentist analysis (i.e., the construction of confidence intervals and bounding the regret when the confidence interval event holds).

2. The parameter $\epsilon$ seems a bit mysterious to me. It seems like it is a parameter that is used only in the analysis. Perhaps the results could be more clearly stated in the main text by just setting $\epsilon$ to the canonical values.

3. The bounds that are obtained are not that easy to interpret.

    a) Any intuition for what the $\xi_a$ functions accomplish (e.g., in line 225)? Can one improve the range for $\xi_a$ to be $\Theta(1/\mathrm{poly}(n))$? A slightly unsatisfying aspect of the claim when one takes $n\to \infty$ is that $1/\sqrt{\log n}$ goes to zero very slowly.

    b) What is the intuition for the $\sum_a \int_{\theta_{-a}}$ term? It seems like this comes from prior analysis. Why does it capture the dependence on the prior?

    c) The linear gaussian bandit bound can be used to recover (a version) of the gaussian MAB bound. Can you discuss this? Is this tight in the relevant problem parameters? If not, why?

4. line 147-148: the claim about the setting where $\sigma_0 \to \infty$ is not clear to me why the complexity term should approach zero. Can this be formalized?

5. Can the ideas be extended beyond Gaussian settings, and/or can they handle some degree of model misspecification (e.g., what happens if the rewards are close to gaussian in some distance metric).

6. Experiments:

     a) Did you tune the confidence widths for the BayesUCB and UCB1 algorithm?
     b) Can confidence bands be plotted?
     c) I think the most interesting part is the comparison between BayesUCB and UCB1. Your theory suggests that BayesUCB improves over UCB1 because it takes into account the prior. Can this be demonstrated more robustly, i.e. for a range of experimental parameters. Also, we would expect this to happen in well-specified settings where the rewards are actually Gaussian. It would be interesting to see if this phenomena is robust and holds for real-world datasets (otherwise, in some sense, it might not be interesting to study BayesUCB...)
    d) Why are your plotted bounds not monotone in $\sigma_0 and \Delta_0$? (Taking a step back, is this something that one should expect?)
    e) In some regimes, your bounds for the linear setting are worse than the established $\sqrt{n}$ bounds. Why is this so? Does this point to some looseness in the analysis?

7) At a high level, why is BayesUCB interesting to study, and how does it compare to other Bayesian approaches, e.g., Thompson sampling? Which should one choose to use in practice?


**Limitations:**

None.

---

> ### Author Rebuttal · Authors · 2023-08-08
>
> We would like to thank the reviewer for valuable feedback and detailed questions. We address your concerns below. If you have any additional concerns, please reach out to us to discuss them.
>
> **Q1: Novelty in techniques**
>
> Please see the **global rebuttal**.
>
> **Q2: Parameter $\varepsilon$**
>
> This is an upper bound on the regret of actions bounded trivially in the bounds, by $\varepsilon n$. We suggest $\varepsilon = 1 / n$, and say it right after Theorems 1 and 5.
>
> **Q3: Bounds are hard to interpret**
>
> **a) Role of $\xi_a$**
>
> The terms $\xi_a$ arise due to the intermediate value theorem for function $h_a$ when the gaps are in $[1 / n, 1 / \sqrt{\log n}]$. Similar terms appear in the analysis of [2] but vanish in the final asymptotic bound. The rate $1 / \sqrt{\log n}$ cannot be changed to $1 / \text{poly}{\log n}$ without increasing dependence on $n$ in other parts of the bound.
>
> **b) How does the complexity term of [2] capture prior dependence?**
>
> The term is indeed not easy to interpret in general. Roughly speaking, it captures the difference between prior means. In a Gaussian bandit with $K = 2$ actions, it has a closed form
>
> $$2 \sqrt{\frac{2}{\pi \sigma_0^2}} \exp[- (\mu_{0, 1} - \mu_{0, 2})^2 / 4 \sigma_0^2]$$
>
> Proposition 3 gives a general upper bound for $K > 2$ actions.
>
> **c) Does the linear bandit bound recover the Gaussian bandit bound?**
>
> The bound is $O(K^2 \mathbb{E}[1 / \Delta_{\min}] \log^2 n)$ in a $K$-armed bandit and thus not tight. This is similar to the frequentist gap-dependent bound of
>
> > Abbasi-Yadkori, Pal, and Szepesvari. Improved algorithms for linear stochastic bandits, 2011.
>
> not being tight in a multi-armed bandit. The reason is that the bound is proved for large action spaces.
>
> **Q4: Complexity term goes to zero as $\sigma_0 \to \infty$**
>
> Each term in the complexity term is a product of factors $\sqrt{c_1 / \sigma_0^2}$ and $\exp[-c_2 / \sigma_0^2]$ for some constants $c_1, c_2 > 0$. As $\sigma_0 \to \infty$, the former factor goes to $0$ while the latter goes to $1$. As a result, the complexity term goes to zero.
>
> **Q5: Extension beyond Gaussians**
>
> Our work can be extended to Bernoulli bandits with factored beta priors (second paragraph in Section 7) as follows. Let $\mathrm{Beta}(\alpha_a, \beta_a)$ be the prior distribution of the mean reward of action $a$ and $N_{t, a, +}$ be the number of positive observations of action $a$ up to round $t$. Then the posterior mean and confidence interval width in Section 3.1 are
>
> $$\hat{\theta}\_{t, a} = \frac{\alpha_a + N_{t, a, +}}{\alpha_a + \beta_a + N_{t, a}}$$
>
> and
>
> $$C_{t, a} = \sqrt{\frac{\log(1 / \delta)}{2 (\alpha_a + \beta_a + N_{t, a} + 1)}}$$
>
> In Corollary 2, the constant at the leading complexity term of action $a$ changes to
>
> $$2 \log(1 / \delta) \log n - (\alpha_a + \beta_a + 1) / (2 \log n)$$
>
> and the bound would have the same interpretation as before. In fact, all theory in Sections 4.1 and 4.2 can be extended similarly.
>
> Theoretically-sound handling of model misspecification in Bayesian algorithms and analyses is generally difficult. We report empirical results for model misspecification in Q6.
>
> **Q6: Experiments**
>
> **a) Are confidence widths tuned?**
>
> No. Implemented as analyzed.
>
> **b) Can you plot confidence bands?**
>
> We believe that you mean standard errors in BayesUCB and UCB1 plots in Figure 1. Note that they are plotted. They are just very narrow due to a high number of runs ($1000$).
>
> **c) More extensive comparison of BayesUCB and UCB1**
>
> See Figure 1 in the attached pdf to the **global rebuttal**. We report the difference in regret of UCB1 and BayesUCB on $81$ Bayesian bandit instances, obtained by all combinations of
>
> * Number of actions $K \in \\{5, 10, 20\\}$
> * Reward noise $\sigma \in \\{0.5, 1, 2\\}$
> * Prior gap $\Delta_0 \in \\{0.5, 1, 2\\}$
> * Prior width $\sigma_0 \in \\{0.5, 1, 2\\}$
>
> In Figure 1a, the noise is Gaussian $\mathcal{N}(0, \sigma^2)$. In Figure 1b, the noise is $\sigma$ with probability $0.5$ and $- \sigma$ otherwise. This noise is $\sigma^2$-sub-Gaussian, the same magnitude as $\mathcal{N}(0, \sigma^2)$ but far from it in terms of the distribution. This tests the robustness of BayesUCB to assuming Gaussian noise in the posterior updates. UCB1 only requires $\sigma^2$-sub-Gaussian noise. All results are averaged over $1000$ runs.
>
> Out of $162 = 2 \cdot 81$ tested instances, UCB1 improves upon BayesUCB only once. In this case, the difference in regret is almost zero and likely due to the randomness of only $1000$ runs. BayesUCB is remarkably robust to noise misspecification, although this is hard to prove.
>
> **d) Are Bayes regret bounds monotone?**
>
> Not necessarily. As an example, the upper bound on the complexity term in Proposition 3 goes to zero when either $\sigma_0 \to 0$ or $\sigma_0 \to \infty$. We discuss this right after Proposition 3.
>
> **e) Our bounds can be worse than prior bounds**
>
> We improve over prior bounds when the complexity term, $1 / \Delta$ in expectation, is small. This allows us to capture easy bandit instances that the prior works did not capture. When $1 / \Delta$ in expectation is high, our bounds can be worse than $\tilde{O}(\sqrt{n})$ bounds. This is analogous to the relation of frequentist gap-dependent and gap-free  bounds in bandits. To get the best bound, the minimum of the two should be taken.
>
> **Q7: Why is BayesUCB interesting to study?**
>
> BayesUCB is empirically competitive with Thompson sampling. This was reported, for instance, in Figures 1, 2, and 4 in
>
> > Kveton, Meshi, Zoghi, and Qin. On the value of prior in online learning to rank, 2022.
>
> It is easier to analyze though because it acts according to an upper confidence bound. Since the submission of this work, we extended the arguments in Section 4.1 to Thompson sampling.
>
> **References**
>
> > [1] Russo and Van Roy. Learning to optimize via posterior sampling, 2014.
>
> > [2] Lai. Adaptive treatment allocation and the multi-armed bandit problem, 1987.

---

> > ### Comment · Reviewer_Q5oQ · 2023-08-14
> > **Thanks for your reply.**
> >
> > *This is analogous to the relation of frequentist gap-dependent and gap-free bounds in bandits.* Can you point me to a reference for this fact?
> >
> > Also, how do your results compare to other works on instance-dependent learning:
> > [1] Andrew Wagenmaker, Dylan J. Foster. "Instance-Optimality in Interactive Decision Making: Toward a Non-Asymptotic Theory."

---

> > > ### Author Response · Authors · 2023-08-14
> > > **Further clarifications**
> > >
> > > Thank you for following up!
> > >
> > > Gap-dependent bounds can indeed be worse than gap-free bounds. See Section 7 (*The Upper Confidence Bound Algorithm*) in
> > >
> > > > Lattimore and Szepesvari. Bandit algorithms, 2020.
> > >
> > > A frequentist gap-dependent bound of UCB1 in a $K$-armed bandit with $1$-sub-Gaussian rewards (Theorem 7.1) is $O(K \Delta^{-1} \log n)$, where $\Delta$ is the minimum gap. A frequentist gap-free bound in the same setting (Theorem 7.2) is $O(\sqrt{K n \log n})$. It follows that the latter is lower than the former when the gap is small, $\Delta = o(\sqrt{(K \log n) / n})$.
> > >
> > > We are aware of
> > >
> > > > Wagenmaker and Foster. Instance-optimality in interactive decision making: Toward a non-asymptotic theory, 2023.
> > >
> > > This is a recent attempt in theory to design general adaptive algorithms with good finite-time instance-dependent bounds based on optimal allocations. The promise of these methods is to be more statistically efficient than optimism. The catch is that they may not be implementable, as discussed in Section 2.2. This work is also in the frequentist setting.
> > >
> > > Finite-time instance-dependent / gap-dependent frequentist regret bounds are very common in bandits, because this is the standard way of analyzing $K$-armed bandits. Our work derives the first comparable finite-time bounds, logarithmic in $n$, in the Bayesian setting. We believe that this is a major progress in theory, similar to first finite-time gap-dependent frequentist regret bounds of
> > >
> > > > Auer, Cesa-Bianchi, and Fischer. Finite-time analysis of the multiarmed bandit problem, 2002.

---

### Official Review · Reviewer_MGeP · 2023-07-03

**Soundness:** 3 good
**Presentation:** 3 good
**Contribution:** 3 good
**Rating:** 6
**Confidence:** 4

**Summary:**

This paper considers a Bayesian setting of the multi-armed bandit problem and provides finite-time regret bounds. For the Gaussian bandits with independent priors, the authors provide regret analyses for BayesUCB and UCB1 algorithms. Both bounds scale as $O(c_h \log^2 n)$ and match the lower bound by Lai (1987) up to constant factors. For the Linear bandits with Gaussian rewards, the authors provide an analysis for BayesUCB and the bound scales as $ O(c_\Delta \text{polylog}n)$. The empirical results corroborate the tightness of the bounds.

**Strengths:**

This paper gives the first finite time bound for the Bayesian regret minimization problem with logarithmic regret for Gaussian reward and Gaussian prior setting. In the case of the multi-armed (unstructured) bandit setting, for both BayesUCB and UCB1, the order of $O(c_h \log^2 n)$ Bayesian regret bound is given. For the linear Gaussian bandit case with bounded inverse gap, the bound does not scale with the number of arms and scales as $O(c_\Delta \text{polylog} n)$.

**Weaknesses:**

Lai (1987) provides an upper bound that exactly matches the lower bound, including the universal constants. The authors' bound does not seem to match it asymptotically.


There is an assumption that the prior is factored. (Lai (1987) has weaker condition on the prior distribution (3.16) - (3.18) in Theorem 3.) I believe this assumption should be more explicitly stated/discussed in the main theorem.

**Questions:**

If possible,

- Could you provide the reasoning/intuition behind why it would be difficult to have matching upper bounds with constants asymptotically together with the finite round bounds?
- Why the factored assumption is necessary or may not be?
- Could you make contrast your proof with the proof of Lai (1987)? What techniques/devices make it possible to bound with a finite sample?
- Why would it be difficult to get a lower bound with the linear bandit setting?
- Would it be possible to get Bayesian regret bound for the linear bandit setting by integrating the complexity term without finite inverse gap  assumption?


**Limitations:**

See the Weaknesses section.

---

> ### Author Rebuttal · Authors · 2023-08-08
>
> We would like to thank the reviewer for valuable feedback. We address your concerns below. If you have any additional concerns, please reach out to us to discuss them.
>
> **Q1: Not all constants in the upper bound match asymptotically**
>
> We believe that you mean $\sigma^2$ in the leading term of Corollary 2. If not, please respond to us and clarify. In our analysis, this term arises due to a high-probability upper bound on the number of times that a suboptimal action is taken, which naturally depends on reward variance $\sigma^2$. In the asymptotic analysis in [2], in the first step of the proof of Theorem 3(i), the authors take a step that roughly translates to $(\theta_i - \theta_j) / [(\theta_i - \theta_j)^2 / 2 \sigma^2] \sim 2 (\theta_i - \theta_j)^{-1}$ in a Gaussian bandit. Thus $\sigma \sim 1$ and our upper bounds would match. We will add a discussion on this to the paper.
>
> **Q2: Is the factored prior assumption necessary?**
>
> The factored assumption is not needed in the algorithm design. This is because BayesUCB in a Gaussian bandit (Section 3.1) can be implemented as BayesUCB in a linear bandit with Gaussian rewards (Section 3.2), where the actions are the standard Euclidean basis.
>
> To obtain the bounds in Section 4.1, we rely on the assumption that the posterior distribution of each action depends only on that action, in Equation 5 in Section 5.1. However, since a Gaussian bandit is a special case of a linear bandit with Gaussian rewards, the logarithmic regret bound in Theorem 5 also applies.
>
> **Q3: Novelty in techniques**
>
> Please see the **global rebuttal**.
>
> **Q4: Lower bound in the linear bandit**
>
> We did not attempt to derive it. The main goal of this paper were finite-time logarithmic Bayes regret upper bounds.
>
> **Q5: Integrating the complexity term in Theorem 5**
>
> This is certainly possible and an interesting direction for future work. We did not do it because the main reason for deriving Corollary 2, an upper bound on Theorem 1 that does not involve $1 / \Delta$, is that it matches an asymptotic lower bound in [2]. No such instance-dependent lower bound exists in Bayesian linear bandits.
>
> **References**
>
> > [1] Russo and Van Roy. Learning to optimize via posterior sampling, 2014.
>
> > [2] Lai. Adaptive treatment allocation and the multi-armed bandit problem, 1987.

---

> > ### Comment · Reviewer_MGeP · 2023-08-17
> > **Re: Rebuttal by Authors**
> >
> > Thank you very much for your rebuttal. I don't have further questions now. Thanks.

---

> > > ### Author Response · Authors · 2023-08-17
> > > **Thanks**
> > >
> > > Thank you for letting us know that our rebuttal answered your questions. If anything else arises later, please let us know.
> > >
> > > Sincerely,
> > >
> > > The authors

---

### Official Review · Reviewer_o1t4 · 2023-07-03

**Soundness:** 3 good
**Presentation:** 3 good
**Contribution:** 2 fair
**Rating:** 5
**Confidence:** 3

**Summary:**

The authors consider the Bayesian bandits problem and provide first finite-time logarithmic regret upper bounds by analyzing the BayesUCB algorithm from a new perspective, departing from previous studies. Their theoretical results demonstrate the significance of the prior. Additionally, the authors showcase the versatility of their analysis techniques by applying them to  linear bandits. Furthermore, they substantiate the tightness of their bounds through numerical evaluations.

**Strengths:**

The authors provide first finite-time logarithmic regret upper bounds for Bayesian bandits, which improve the existing $\tilde{O}(\sqrt{n})$ bounds and highlight the significance of the prior.
The authors provide experimental results for both Gaussian bandits and linear bandits with Gaussian rewards.
The experimental results substantiate the theoretical findings.
The results and the addressed problem hold substantial significance and are likely to capture the interest of a broad audience in Neurips.

**Weaknesses:**

I have two primary concerns regarding this work and would like to request the authors to provide additional clarifications.

==Weakness 1==

While the authors present finite-time logarithmic regret upper bounds for Bayesian bandits, it is worth noting that Lai [1987] provide both asymptotic logarithmic upper and lower bounds. Therefore, it would be beneficial for the authors to explicitly highlight the key challenges associated with analyzing the finite-time case. This clarification would help readers better understand the distinctive aspects of the two settings.

==Weakness 2==

From a theoretical standpoint, the authors demonstrate the tightness of their results by comparing them with the asymptotic lower bound presented in Lai [1987]. This may raise concerns about the rigors. Additionally, it introduces a logical paradox concerning the contribution of the paper. If the theoretical analysis between the finite-time and asymptotic cases significantly differs, the comparison may not be sufficiently robust. On the other hand, if the comparison holds, it might suggest that the current results have somewhat limited contributions.

==Minor Comment==

Regarding the title, I suggest revising it to "Finite-Time Logarithmic Bayes Regret Upper Bounds" instead of the current title "Logarithmic Bayes Regret Bounds." This change may be appropriate because Lai [1987] already covers both asymptotic logarithmic upper and lower bounds for Bayesian bandits.

Reference

Tze Leung Lai. Adaptive treatment allocation and the multi-armed bandit problem. The Annals of Statistics, 15(3):1091–1114, 1987.

**Questions:**

Please see the points mentioned in the weaknesses above and provide comments on them.

**Limitations:**

I don't see any issue on societal impact.

---

> ### Author Rebuttal · Authors · 2023-08-08
>
> We would like to thank the reviewer for valuable feedback and noting the significance of our results. We address your concerns below. If you have any additional concerns, please reach out to us to discuss them.
>
> **Q1: Key challenges in analyzing the finite-time case**
>
> Please see the **global rebuttal**.
>
> **Q2: Finite-time regret upper bounds are compared to asymptotic regret lower bounds**
>
> In frequentist bandit analyses, it is standard to compare asymptotic lower bounds to finite-time upper bounds because finite-time logarithmic lower bounds do not exist. This is why we did not stress this particular point. Using the same reasoning, our discussion after Corollary 2 shows that the Bayes regret of BayesUCB is order optimal as $n \to \infty$.
>
> That being said, our finite-time upper bounds also reveal an interesting difference from the asymptotic lower bound in [2], which deserves more future attention. Specifically, the regret bound of BayesUCB (Bayesian algorithm in Corollary 2) improves with prior information given to the algorithm while that of UCB1 (frequentist algorithm in Theorem 4) does not. We observe these improvements empirically as well (Section 6). However, both bounds are asymptotically order-optimal when compared to the lower bound in [2] (Section 4.2) because the benefit of knowing the prior vanishes in asymptotic analyses. This motivates the need for finite-time logarithmic Bayes regret lower bounds, which do not exist.
>
> **Make paper title more precise:** Agreed.
>
> **References**
>
> > [1] Russo and Van Roy. Learning to optimize via posterior sampling, 2014.
>
> > [2] Lai. Adaptive treatment allocation and the multi-armed bandit problem, 1987.

---

> > ### Comment · Reviewer_o1t4 · 2023-08-17
> > **Response to Rebuttal**
> >
> > Thank you for your detailed responses to my questions. I have no further questions at this point.

---

> > > ### Author Response · Authors · 2023-08-17
> > > **Thanks**
> > >
> > > Thank you for acknowledging that we answered your questions. If anything else arises later, please let us know.
> > >
> > > Sincerely,
> > >
> > > The authors

---

### Official Review · Reviewer_H2AN · 2023-07-14

**Soundness:** 3 good
**Presentation:** 3 good
**Contribution:** 3 good
**Rating:** 6
**Confidence:** 3

**Summary:**

This paper derives a finite-time regret bound for BayesUCB in Bayesian bandits. In Gaussian bandits, the bound grows linearly in $c_h \log^2(n)$, where $n$ is the time horizon, and $c_h$ is a constant determined by the prior. The bound matches that given by Lai (1987). A regret bound of BayesUCB in linear bandit is also established. Numerical experiments have also been conducted to support the theoretical findings.

**Strengths:**

The paper studies an important problem and derives useful regret bounds. The paper is also well-organized and easy to follow.


**Weaknesses:**

In terms of presentation/ writing, I wish the paper could be more polished. Some of the examples are listed in “Questions.”


**Questions:**

There are some minor issues in writing. Below are a couple examples:
1. In line 57, the paper reads “$A_t \in \mathcal{A}$.” However, $A_t$ is a random variable taking values in $\mathcal{A}$, so the statement is not precise.
2. In line 83, it reads” $C_{t,a}$ is a confidence interval.” However, $C_{t,a}$ should be half the length of the confidence interval.
3. Typo in line 97: should remove “a” from “the posterior distribution is a Gaussian.”

---

> ### Author Rebuttal · Authors · 2023-08-08
>
> We would like to thank the reviewer for appreciating the usefulness of our results and their writing suggestions. $A_t$ is indeed a random variable with values in $\mathcal{A}$. The reviewer is also right that $C_{t, a}$ is a half of the width of the confidence interval centered at the posterior mean.
>
> If you have any additional concerns, please reach out to us to discuss them.

---

### Official Review · Reviewer_7MLE · 2023-07-23

**Soundness:** 3 good
**Presentation:** 3 good
**Contribution:** 3 good
**Rating:** 6
**Confidence:** 1

**Summary:**

The paper studies the multi-arm bandit (MAB) problem, which is an online learning problem where a learner interacts with an environment over multiple rounds. In each round the learner would make an action, and each different action will incur a reward sampled from some distribution parametrized by an unknown $\theta$. The goal is to get highest expected cumulative reward overtime, or in other words, to minimize the cumulative "regret" of not choosing the best action in each round. Here "best" means an action incurring the highest expected reward. The paper focuses on Bayesian MAB, where the unknown parameter $\theta$ itself is generated from some known prior.

The paper proves several improved regret bounds for various Bayesian MAB settings, including: proving finite-time logarithmic Bayesian regret bounds for a Bayesian UCB algorithm in a Gaussian bandit, deriving a finite-time logarithmic Bayesian regret bound for a frequentist UCB algorithm to demonstrate the value of the prior as side information, proving a Bayesian regret bound for a Bayesian linear bandit algorithm, and empirically comparing the tightness of the bounds to prior bounds.


**Strengths:**

The authors provide a logarithmic regret bounds for some basic Bayesian MAB problems, which improves exponentially on previous best result and matches an asymptotic lower bound. This seems to me a breakthrough.


**Weaknesses:**

N/A

**Questions:**

First, The paper under consideration falls outside my domain and I never bid for it (Regrettably I wasn't given any alternative submissions to assess). Despite my limited familiarity with MAB, the problems delineated in the manuscript, such as the Gaussian bandits, seem both intuitive and natural. The lower bound result referenced, by Lai et al., dates back several decades. Given this context, I presume these matters have been thoroughly examined within the domain.

Given this, one thing that appears perplexing to me is the paper's claim of exponential improvements over prior best results. Such a significant leap over such basic problems, usually suggests the introduction of some truly innovative ideas in the analysis. However, the proof seems quite innocuous and it's not immediately clear to me what is the core idea that leads to the improvement.

It will be very helpful if the authors can add one or two paragraphs elucidating previous results. Specifically, what is the major barrier that prevents them from achieving this logarithmic bound, and how the authors manages to overcome that barrier.

---

> ### Author Rebuttal · Authors · 2023-08-08
>
> We would like to thank the reviewer for valuable feedback and appreciating the significance of our results. For technical novelty, please see the **global rebuttal**. If you have any additional concerns, please reach out to us to discuss them.
>
> We also wanted to add the following note. We improve over prior bounds when the complexity term, $1 / \Delta$ in expectation, is small. This allows us to capture easy bandit instances that the prior works did not capture. When $1 / \Delta$ in expectation is high, our bounds can be worse than $\tilde{O}(\sqrt{n})$ bounds. This is analogous to the relation of frequentist gap-dependent and gap-free  bounds in bandits. To get the best bound, the minimum of the two should be taken.

---

### Author Rebuttal · Authors · 2023-08-08

We would like to thank all reviewers for feedback, and appreciating the novelty of our work and its potential impact. One comment that appeared in most reviews is that the technical novelty of the work was never clearly presented in one place. We summarize it below and address all other comments in your respective rebuttals.

**Technical novelty**

All modern Bayesian analyses follow the work of [1], which derived the first finite-time $\tilde{O}(\sqrt{n})$ Bayes regret bounds for BayesUCB and Thompson sampling. The key idea in these analyses is that conditioned on the history, the optimal and taken actions are identically distributed, and that upper confidence bounds are deterministic functions of the history. This is the main step where the randomness of instances in Bayesian bandits is used. Based on this, the regret in round $t$ is bounded by the confidence interval width of the taken action, and the usual $\tilde{O}(\sqrt{n})$ bounds can be obtained by summing up the confidence interval widths over $n$ rounds.

In our work, we first bound the regret in a fixed bandit instance, similarly to existing frequentist analyses. The bound involves $1 / \Delta$ and is derived using biased Bayesian confidence intervals. The rest of our analysis is Bayesian in two parts: we (1) prove that the confidence intervals fail with a low probability and (2) bound random $1 / \Delta$, following a similar technique to [2]. The resulting logarithmic Bayes regret bounds cannot be derived using the techniques in [1], as these become loose when the confidence interval widths are introduced.

Asymptotic logarithmic Bayes regret bounds were derived in [2]. From this analysis, we only reuse the technique for bounding $1 / \Delta$ in expectation, when proving Corollary 2 to Theorem 1. The central part of our proof is a finite-time per-instance bound on the number of times that a suboptimal action is taken. A comparable argument in [2], in their Theorem 2, is an asymptotic bound on average over the bandit instances. To show the generality of our technique, we also apply it to a structured linear bandit problem.

**References**

> [1] Russo and Van Roy. Learning to optimize via posterior sampling, 2014.

> [2] Lai. Adaptive treatment allocation and the multi-armed bandit problem, 1987.

---

### Decision · Program_Chairs · 2023-09-21

**Decision:**

Accept (poster)

**Comment:**

Gap dependent bounds for bandits have been known for decades. This paper shows that a Bayesian algorithm with slightly biased estimates can enjoy finite time gap dependent bounds like those first proved for UCB. The reviewers all like the paper and the contribution is meaningful and novel. But, at the same time, the contribution isn't ground breaking and is not likely to lead to any new practical recommendations for users of bandit algorithms.